# TruePrime is a novel method for whole-genome amplification from single cells based on *Tth*PrimPol

Ángel J. Picher[1], Bettina Budeus[2], Oliver Wafzig[2], Carola Krüger[2], Sara García-Gómez[3],
María I. Martínez-Jiménez[3], Alberto Díaz-Talavera[3], Daniela Weber[2], Luis Blanco[3] & Armin Schneider[2]

Sequencing of a single-cell genome requires DNA amplification, a process prone to introducing bias and errors into the amplified genome. Here we introduce a novel multiple displacement amplification (MDA) method based on the unique DNA primase features of *Thermus thermophilus* (*Tth*) PrimPol. *Tth*PrimPol displays a potent primase activity preferring dNTPs as substrates unlike conventional primases. A combination of *Tth*PrimPol's unique ability to synthesize DNA primers with the highly processive Phi29 DNA polymerase (Φ29DNApol) enables near-complete whole genome amplification from single cells. This novel method demonstrates superior breadth and evenness of genome coverage, high reproducibility, excellent single-nucleotide variant (SNV) detection rates with low allelic dropout (ADO) and low chimera formation as exemplified by sequencing HEK293 cells. Moreover, copy number variant (CNV) calling yields superior results compared with random primer-based MDA methods. The advantages of this method, which we named TruePrime, promise to facilitate and improve single-cell genomic analysis.

[1] SYGNIS Biotech S.L.U., Parque Científico de Madrid, Cantoblanco, Madrid 28049, Spain. [2] SYGNIS Bioscience GmbH, Waldhofer Strasse 104, Heidelberg 69123, Germany. [3] Centro de Biología Molecular Severo Ochoa (CSIC-UAM), Cantoblanco, Madrid 28049, Spain. Correspondence and requests for materials should be addressed to L.B. (email: lblanco@cbm.csic.es) or to A.S. (email: aschneider@sygnis.de or schneider@sygnis.de).

It has become apparent in the last 5 years that genomic analysis of single cells provides crucial information that is lost in bulk sequencing of tissue because of averaging effects and limitations of computational methods to deconvolute sequence information from many different clones[1,2]. For example, the complexity of alterations of the cancer genome has only been grasped recently by assessing single tumour cells in parallel[3–5]. Especially for oncology, single-cell sequencing offers novel insights into the evolution of cancers over time and in reaction to treatment, which will lead to novel strategies for treatment regimens and drug development[6]. Other application areas for single-cell sequencing are pre-implantation diagnostics[7–9] and basic biological research, for example, in the neurosciences[10].

Amplification of genomic DNA is a necessary first step for the available sequencing technologies. Unfortunately, DNA amplification is a process subject to bias introduction, error and co-amplification of minute levels of contaminating DNA. Several techniques have been developed for whole genome amplification (WGA), broadly dividable into PCR-related protocols and those based on multiple displacement amplification (MDA). PCR-based methods can be classified into degenerate oligonucleotide-primed PCR (DOP-PCR)[11], linker-adapter PCR[12], primer extension pre-amplification PCR (PEP-PCR-/I-PEP-PCR)[13,14] and variations thereof. MDA methods are mainly based on using the highly processive Φ29DNApol[15] together with random hexamers[16–19]. There is another variant MDA method called pWGA based on the reconstituted T7 replication system[20]. Recently, another hybrid PCR/MDA method called multiple annealing and looping-based amplification cycles (MALBAC) has been proposed, relying on the *Bacillus stearothermophilus* polymerase for the MDA process[21]. Key parameters that determine the quality of the amplification are the absence of contaminations and artefacts in the reaction products, coverage breadth and uniformity, nucleotide error rates and the ability to recover single-nucleotide variants (SNVs), copy number variants (CNVs) and structural variants. In general, PCR-based methods are thought to have advantages in CNV detectability[22], whereas Φ29DNApol-based methods have the advantage of extremely low nucleotide error rates due to the high fidelity of the polymerase, produce very long amplification products and cover the genome more completely. Problems that affect all amplification methods to some degree are chimera formation and preferential amplification of one allele (allelic dropout, ADO).

A source for potential amplification bias in the current Φ29DNApol-based MDA methods is the propensity to generate primer-derived artefacts and priming inequality arising from different sequence-dependent hybridization kinetics of the oligonucleotides. Thus, using a dedicated primase may provide an advantage over random oligonucleotides. However, most known primases only accept NTPs and generate RNA primers. These RNA primers are not an ideal substrate for most replicative DNA polymerases and need to be elongated by specialized transition DNA polymerases, as DNA polymerase-α in human cells.

Primases can be divided into two evolutionarily unrelated families: DnaG-like primases (Bacteria) and archaeal-eukaryotic primase (AEP)-like primases (Archaea and Eukaryotes)[23,24]. Recently, a novel subfamily of AEPs called primase-polymerase (PrimPol)[25,26] has been described, whose first members were originally found in archaeal plasmids[27] and in some bacteria[28]. PrimPols show both DNA polymerase and DNA primase activities, and are often associated to helicases, to form a replication initiation complex[26,28,29]. These features enable a system where the same enzyme performs both the initiation and elongation stages. Perhaps the most significant feature of PrimPols, unlike conventional primases, is their ability to carry out the initiation and extension of DNA chains[27,30–32].

More recently, PrimPol was described to exist in human cells (HsPrimPol, UniProtKB Q96LW4), encoded by the *PRIMPOL* gene (also known as *CCDC111*)[33–36].

In this case, HsPrimPol is not associated to a helicase, but displays some strand-displacement capacity. Moreover, its DNA polymerase activity is able to efficiently bypass different kind of DNA lesions as 8oxoG and pyrimidine dimers[34,35,37–39]. It is very likely to be that this translesion synthesis capacity is crucial for the demonstrated role of human PrimPol in mitochondrial DNA maintenance[34]. Moreover, the most significant feature of human PrimPol (also extending to archaeal and bacterial PrimPols) is the ability to initiate the synthesis of DNA chains (as a DNA primase), unlike conventional primases that need NTPs to make primers[26–28,34]. Such convenient capacity was shown to be required to re-prime arrested replication forks during nuclear DNA replication in human cells[35,36,40] and also confirmed in avian cells[41].

From a biotechnological perspective, the unique ability to synthesize DNA primers could make PrimPol a useful partner in an MDA-type process. Here we describe the cloning and characterization of the *Thermus thermophilus* PrimPol enzyme and the creation of a novel primer-free WGA method with specific advantages for single-cell genome amplification, which we termed TruePrime.

## Results

**TthPrimPol is a DNA primase with wide template specificity.** Human PrimPol was initially considered as a good candidate for DNA amplification processes. However, the human protein was promptly discarded mainly due to stability issues, probably related to the presence of a Zn-finger domain at its carboxy terminus and also due to its strong dependence on $Mn^{2+}$ ions to activate its priming function[34]. We therefore sought to identify a more stable bacterial orthologue, with optimal enzymatic properties to be exploited for biotechnological applications.

Search of the non-redundant database of protein sequences (National Center for Biotechnology Information, NIH, Bethesda), performed using the BLASTP programme[42] revealed that the hypothetical conserved protein AAS81004.1 (291 amino acids) from *T. thermophilus* strain HB27 contained a PrimPol domain of the type found in bifunctional replicases from archaeal plasmids. Although conventional AEP-like primases, as human Prim1, have the three conserved motifs (A, B and C) that form the primase active site, the PrimPols already characterized have the same three conserved motifs but also a Zn-finger domain required for the DNA primase activity[34–36]. Strikingly, the putative *T. thermophilus* PrimPol contains only the three motifs but no Zn-finger domain (Fig. 1a). Instead, *Tth*PrimPol contains an α-helical PriCT-1 domain, found at the C-terminal of some AEP primases[24]. A detailed amino acid sequence alignment (Fig. 2) of *Tth*PrimPol with its closest bacterial relatives and also with the well-characterized pRN1 PrimPol, and some representatives of other AEP-like members found in archaea, bacteria, phages and plasmids, supports the correct identification of PrimPol in *T. thermophilus* (see legend to Fig. 2 for further details). A more extensive search of the closest *Tth*PrimPol orthologues was carried out, supporting their conservation and monophyletic origin in Bacteria (Supplementary Fig. 1). Moreover, the significant amino acid sequence similarity with pRN1 PrimPol and also with the polymerization domain (PolDom) of *Mycobacterium tuberculosis* LigD (not shown), whose three-dimensional (3D) structures have been solved[26,43,44], was sufficient to generate a 3D model for *Tth*PrimPol in complex with DNA and nucleotide substrates (Fig. 1b; see Methods for details).

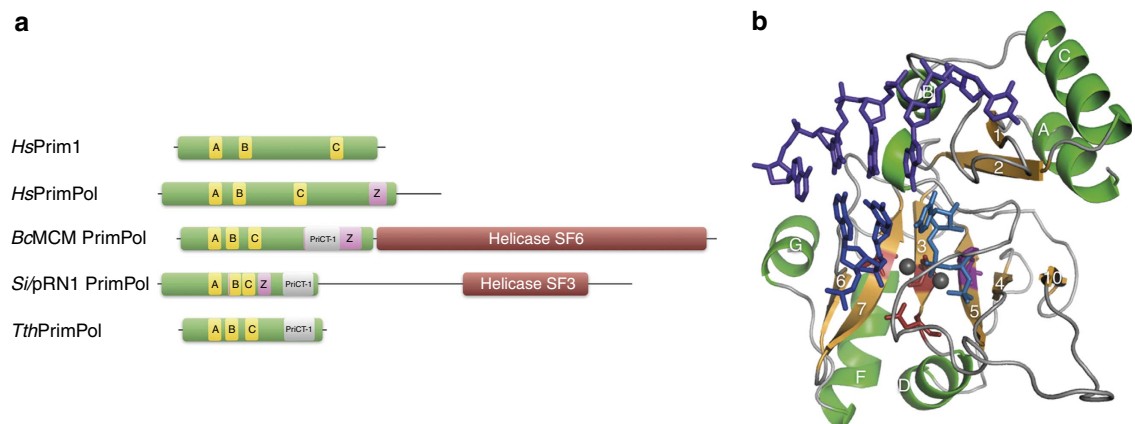

**Figure 1 | A putative PrimPol in *T. thermophilus*.** (**a**) Modular organization of various AEP-like enzymes. A conserved AEP domain (green bar) contains the three conserved regions A, B and C forming the primase active site. Unlike conventional primases as *Hs*Prim1, PrimPols frequently have a Zn-finger-containing region (*Hs*PrimPol) or even a helicase domain (*Bc*MCM PrimPol; *Si*/pRN1 PrimPol). A putative AEP-like enzyme in *T. thermophilus* lacks both Zn finger and helicase domain; however, its C-terminal domain contains a PriCT-1 domain characteristic of some prokaryotic primases, also shared by *Bc*MCM and *Si*/pRN1 PrimPols (see later in **b**). Nomenclature: small catalytic subunit of the human RNA primase (*Hs*Prim1); human PrimPol (*Hs*PrimPol); PrimPol-helicase from *Bacillus cereus* (*Bc*MCM); plasmid pRN1 ORF904 from *Sulfolobus islandicus* (*Si*/pRN1 PrimPol); putative PrimPol from *T. thermophilus* (*Tth*PrimPol). (**b**) 3D structure of *Tth*PrimPol. The computer-modelled crystal structure of *Tth*PrimPol (amino acids 4–208 modelled as described in Methods) is depicted in ribbon format by using the graphic program PyMol. α-Helices are green (lettered), β-strands are orange (numbered) and intervening loop regions are grey; metal ligands (Asp70, Asp72 and Asp123) are shown in red; dNTP ligand (His101) is shown in purple; DNA template (dark purple) and primer (blue) strands, activating metals (grey spheres) and incoming nucleotide (cyan) are derived from 3D structures of *M. tuberculosis* PolDom Ligase D (4MKY and 3PKY).

*Tth*PrimPol was cloned, expressed and purified in a soluble and active form, as described in Methods. Primase activity was first analysed at 55 °C using a single-stranded template oligonucleotide in which a potential primase recognition sequence 3′-**GTC**C-5′ is flanked by thymine residues[34], according to the preferred template context to initiate primer synthesis by several viral, prokaryotic and eukaryotic RNA primases[34–36]. *Tth*PrimPol displayed a strong primase activity, starting synthesis opposite the '**TC**' template sequence. The nucleotide acting as 'nano-primer' (5′-position)[38] can be either a ribonucleotide (ATP) or a deoxynucleotide (dATP) in the presence of manganese, but only a deoxynucleotide (dATP) when magnesium is the metal cofactor (Fig. 3a, left panel). Second and further added nucleotides (3′-position) must be strictly deoxynucleotides (dGTP and dATP), regardless of the metal cofactor present. Modification of the base preceding the directing TC template sequence had a minor effect on the priming activity of *Tth*PrimPol (Fig. 3a, right panel), in contrast to the strong preference for 3′-GTCC-5′ shown by human PrimPol[34]. *Tth*PrimPol was able to initiate DNA primer synthesis also at 30 °C at multiple sites on a single-stranded circular DNA template (M13mp18) by using both purine and pyrimidine nucleotides to form the initial dimer (Fig. 3b), in agreement with a desirable and wide template specificity, unlike human PrimPol that largely prefers to make dimers with purine nucleotides[34]. The preference for dNTPs as incoming nucleotides also makes *Tth*PrimPol a competent DNA-directed DNA polymerase, able to extend the initiating dimers into longer DNA primers. By providing the four dNTPs, *Tth*PrimPol synthesized primers up to 20-mer (Fig. 3c, lane 3). Heparin, in an amount that can inhibit *Tth*PrimPol when pre-incubated (Fig. 3c, lane 2), was only able to inhibit the synthesis of primers longer than 10 nucleotides, when added after enzyme/DNA binding (Fig. 3c, lane 4). Thus, the main primer products (7–9 nt) are synthesized processively, but further extension appears to be distributive.

**Tth*PrimPol serves as primase for Φ29DNApol-mediated MDA.*** Having established that *Tth*PrimPol is indeed a DNA primase,

we explored the possibility that these DNA primers could be efficiently elongated by a second polymerase, the high-fidelity Φ29DNApol[45]. We designed a first experiment in two steps (pulse and chase), to interrogate about the size of the primers made by *Tth*PrimPol that can be efficiently extended by Φ29DNApol. First, during the pulse, *Tth*PrimPol generated labelled primers at different enzyme/DNA ratios, supporting that 7–9 nt primers are the main products, processively synthesized (Fig. 3d, left panel). During the chase at 10 μM dNTPs, Φ29DNApol was able to generate highly elongated products by extending these primers (Fig. 3d, right panel).

Thus, the compatibility of both enzymes allowed their combination to perform rolling circle amplification (RCA)[46] on a single-stranded M13mp18 template, as an alternative method (that we termed TruePrime) to the currently used mix of random primers (RPs) and Φ29DNApol (Fig. 4a). As a control, none of the enzymes by itself was able to amplify the target. Of note, also human PrimPol was not able to cooperate with Φ29DNApol in RCA, highlighting the unique primase features of *Tth*PrimPol, able to make DNA primers in the presence of Mg$^{2+}$, a metal needed to achieve faithful DNA synthesis by Φ29DNApol. In addition, WGA from human DNA yielded DNA amounts comparable to RP-mediated amplification (Fig. 4b). The size distribution of the resulting DNA showed a broad high-molecular weight pattern as also seen with RPs + Φ29DNApol (Fig. 4c). Sensitivity of TruePrime for low-input amounts of target DNA was excellent, in the range of femtograms, and superior to the RP-mediated amplification (Fig. 4d). The reaction output of the *Tth*PrimPol/Φ29DNApol combination was shown to be target derived: even at 1 fg input >95% of the sequences could be mapped to the human genome (Supplementary Fig. 2).

**TruePrime WGA yields high-quality genomic sequences.** We next applied this novel WGA method to the amplification of genomic DNA from single human HEK293 cells isolated by serial dilution, manual picking and visual inspection, and subjected them to the TruePrime protocol (see Methods) for 3 or 6 h reaction time. Yields obtained were ∼6 μg for 3 h and ∼10 μg for

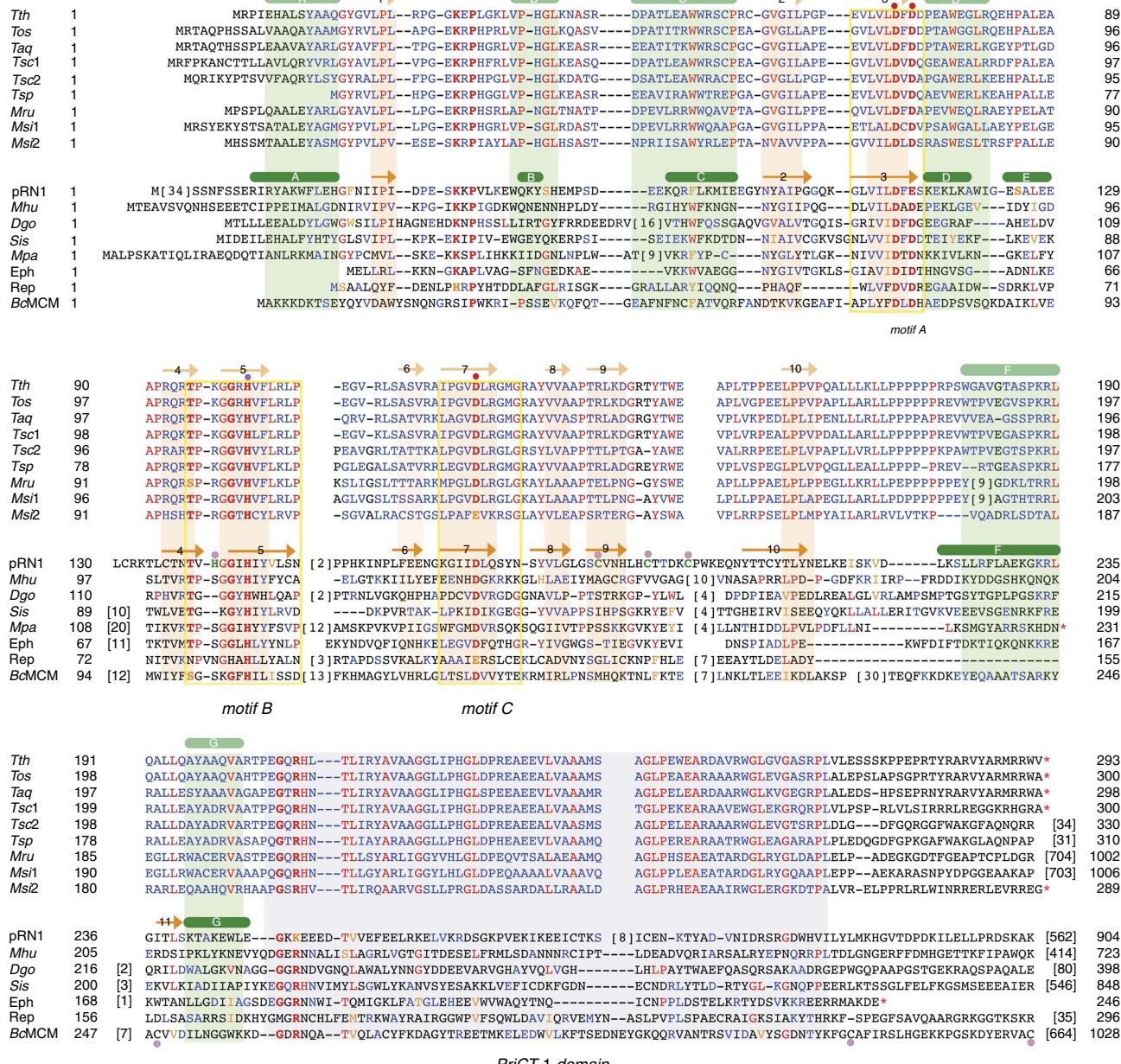

**Figure 2 | Multiple amino acid sequence alignment of the closest *Tth*PrimPol orthologues.** The first block of sequences corresponds to Thermales (*Thermus* and *Meiothermus*) and the second block includes *Si*/pRN1 PrimPol and some other putative bifunctional primases/polymerases from bacteria, archaea and phage; in addition, plasmidic Rep (a potential PrimPol) and *Bc*MCM PrimPol were included. Numbers in parentheses indicate the number of amino acid residues not shown. Invariant or conserved residues among Thermales (first set of sequences) were labelled in red and blue letters, respectively. Identity matches in the second set of sequences were equally coloured. The alignment defines several conserved regions, including the highly conserved motifs A, B and C (boxed in yellow), characteristic of AEP-like primases. Experimentally determined secondary structure elements in *Si*/pRN1 PrimPol are indicated above pRN1 sequence (α-helices, lettered cylinders; green) and β-strands (numbered arrows; orange). Modelled secondary structure elements in *Tth*PrimPol are tentatively depicted above the *Tth*PrimPol sequence (see also Fig 1b). The corresponding aligned regions are boxed in the same colours to emphasize structural conservation between *Tth*PrimPol and the AEP core of pRN1. The C-terminal region of *Tth*PrimPol, conserved in other Thermales (boxed in grey), aligns with the PriCT-1 domain of Rep and Eph primases; this region is not yet crystallized in pRN1 and it has been described as pRN1_helical[29]. Dots indicate invariant residues acting either as metal (red), nucleotide (purple) or Zn (magenta) ligands. Selection of the closest *Tth*PrimPol homologues and multiple alignment of their amino acid sequence were initially performed with the BLAST programme and further adjusted manually to maximize similarities with the structured regions of pRN1 PrimPol. Nomenclature: YP_004631.1 *T. thermophilus* (*Tth*); YP_006971229.1 *Thermus oshimai* (*Tos*); WP_003046664.1 *Thermus aquaticus* (*Taq*); YP_004202830.1 *Thermus scotoductus* (*Tsc*1); YP_004202855.1 *T. scotoductus* (*Tsc*2); ETN89075.1 *Thermus sp* (*Tsp*); YP_003508539.1 *Meiothermus ruber* (*Mru*); YP_003684976.1 *Meiothermus silvanus* (*Msi*1); YP_003684747.1 *M. silvanus* (*Msi*2); AAC44111.1 plasmid pRN1 ORF904 from *S. islandicus* (pRN1); YP_502469.1 *Methanospirillum hungatei* (*Mhu*); YP_006262572.1 *Deinococcus gobiensis* (*Dgo*); YP_002829910.1 *S. islandicus* (*Sis*); YP_003357218.1 *Methanocella paludícola* (*Mpa*); AFO10831.1 *Enterococcus* phage EfaCPT1 (Eph); YP_009074444.1 *Shigella sonnei* Rep protein from plasmid ColE4-CT9 (Rep); WP_044797243 PrimPol-helicase from *B. cereus* (*Bc*MCM).

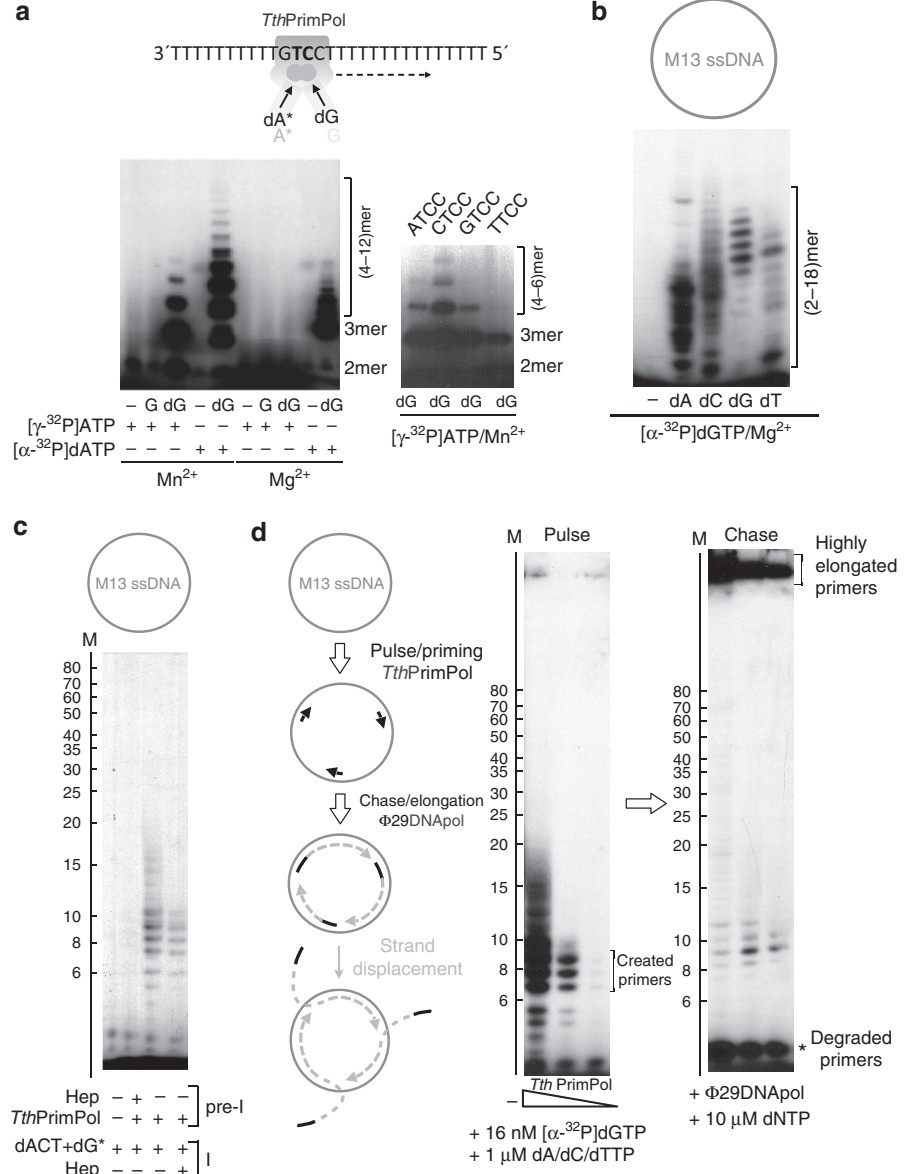

**Figure 3 | _Tth_PrimPol is a DNA primase that can be coupled to processive elongation by Φ29DNApol.** (**a**) Left panel: metal and sugar selectivity of _Tth_PrimPol primase activity. 3′-GTCC-5′ oligonucleotide (1 μM) was used as a preferred template. Labelled nucleotide [γ-$^{32}$P] ATP or [α-$^{32}$P] dATP nM) were alternatively used as 5′-nucleotide and either unlabelled GTP or dGTP (10 μM) were tested as 3′-nucleotide to form the initiating dimer. Primer synthesis mediated by _Tth_PrimPol (400 nM) was evaluated either with 1 mM MnCl$_2$ or 5 mM MgCl$_2$ at 55 °C during 60 min. Right panel: recognition of the priming site. The assay was as in the left panel, but using templates differing in the base preceding the primase initiation site (…X**TC**C…) and the indicated metal and nucleotides. (**b**) _Tth_PrimPol-mediated DNA primer synthesis at 30 °C occurs at multiple sites on a heterogeneous ssDNA template. _Tth_PrimPol (100 nM) was able to generate DNA primers on circular M13mp18 ssDNA (5 ng μl$^{-1}$), when using four alternative combinations of dNTPs, implying that initiation occurred at multiple sites. In all cases, [α-$^{32}$P] dGTP (16 nM) was provided to label the nascent primers, combined with either dATP, dCTP, dGTP or dTTP (1 μM), in the presence of 10 mM MgCl$_2$ at 30 °C during 20 min. (**c**) To evaluate the processivity of primer synthesis by _Tth_PrimPol, we used heparin as a competitor. _Tth_PrimPol (10 nM) was preincubated for 5 min on ice, either in the absence/presence of heparin (1 ng μl$^{-1}$). Subsequently, the reaction was complemented with M13mp18 ssDNA (5 ng μl$^{-1}$), dATP, dCTP and dTTP (10 μM each), [α-$^{32}$P] dGTP (16 nM; 3,000 Ci mmol$^{-1}$) and heparin (1 ng μl$^{-1}$) when indicated and the incubation was maintained for 10 min at 30 °C, and processed as described. (**d**) _Tth_PrimPol-synthesized DNA primers are efficiently extended by Φ29DNApol. The contribution of each enzyme was assayed in two consecutive stages (pulse and chase), as indicated in the scheme. The pulse demonstrated the synthesis of primers with a mean size of 7–9 nt (left panel). During chase (right panel), Φ29DNApol generated high-molecular-weight primer-elongated products (detected at the top of the gel) and some degradation products (evidenced at the bottom).

6 h as quantified by Picogreen. Limited sequencing of DNA obtained from four cells was carried out in comparison with DNA isolated from the originating bulk cells (non-amplified, NA). In parallel, single cells were amplified using a commercially available MDA kit (REPLI-g Single Cell Kit, Qiagen, Aarhus, Denmark), or by using our TruePrime protocol, but exchanging _Tth_PrimPol

for random hexamers (generic RP-MDA). In addition, we amplified a single HEK293 cell by the MALBAC protocol[21] as a hybrid MDA method. Fragment sizes for the amplified DNA were ∼9–19 kb for the commercial RP-MDA, 1.5–12 kb for TruePrime and 0.5–1.5 kb for MALBAC (Supplementary Fig. 3). DNA was sequenced using a paired read strategy (Illumina HiSeq, 125 bp read length).

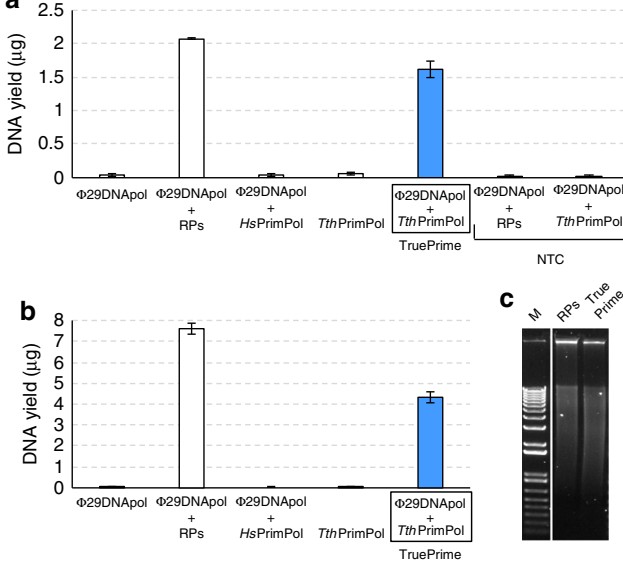

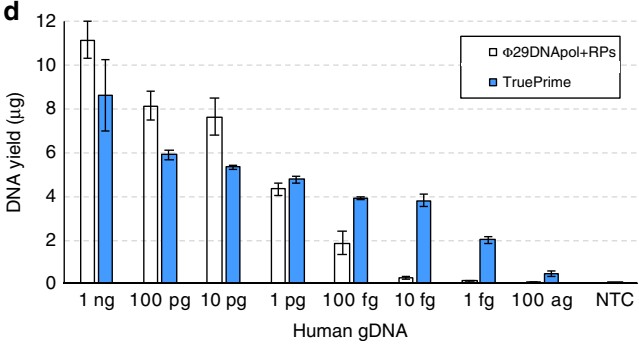

**Figure 4 | *Tth*PrimPol-mediated MDA (TruePrime) is able to efficiently amplify circular and linear DNA molecules.** (**a**) *Tth*PrimPol can be used instead of RPs to trigger MDA by Φ29DNApol. Combination of both enzymes (TruePrime) is able to proficiently amplify single-stranded M13mp1 circular DNA (100 fg), unlike the combination of Φ29DNApol plus human PrimPol. Individual addition of Φ29DNApol or *Tth*PrimPol does not lead to amplification of input DNA. Non-template controls (NTC) are included, to ensure the lack of background amplification in the absence of input DNA. Error bars are s.d. (**b**) Amplification of human genomic DNA (6 pg; the equivalent amount contained in a single human cell) by combination of *Tth*PrimPol and Φ29DNApol. *Hs*PrimPol again failed to amplify DNA in cooperation with Φ29DNApol, highlighting the requirements of the specific primase features of *Tth*PrimPol. Error bars are s.d. (**c**) Agarose gel image showing the similar high size distribution of the amplified fragments obtained with TruePrime versus RPs in part B. (**d**) Superior sensitivity of *Tth*PrimPol-mediated MDA (TruePrime) for the amplification of human genomic DNA (efficient with a DNA input as low as 1 fg), having about 100-fold higher sensitivity than RP-mediated MDA. Error bars are s.d.

Comparison of mapping characteristics of these samples was done at exactly 12 million randomly selected read pairs for NA DNA, TruePrime-amplified DNA, the two RP-MDA protocols and MALBAC. We calculated the deviation of the actual fraction of the human genome covered from the theoretically possible fraction covered assuming an ideal Poisson distribution for the successfully mapped reads[47]. Theoretically expected coverage rates varied because of differing success rates in mapping the 12 million read pairs to the genome (NA DNA 92.03%, TruePrime 86.77%, Commercial random primed MDA 91.50%,

Generic random primed MDA 59.07%, MALBACs 89.68%). In addition, we adjusted the expected maximal coverage to the duplicate rate, which was 1.79% in the NA sample, 1.23% in the TruePrime sample, 1.33% in the commercial RP-MDA sample, 9.53% in the generic RP-MDA sample and 0.23% for MALBAC. The deviations from the observed to the maximally expected (Poisson) coverage breadth were 9.17% for NA DNA, 13.15% for TruePrime, 34.83% for the commercial RP-MDA, 30.73% for the generic RP-MDA and 46.89% for MALBAC at this read depth. Visual inspection of the coverage pattern across the genome by a Circos plot (Fig. 5a) as well as by a sliding window view on chromosome 4 (Fig. 5b and Supplementary Fig. 4) highlights the evenness of coverage and the high similarity to the NA material in contrast to the two RP-MDA methods and MALBAC. Graphing read depth frequency also shows the similarity of the TruePrime amplified sample to the NA one (Fig. 5c).

We studied terminal breadth of coverage in one of the amplified genomic samples (1c), the commercial RP-MDA sample, the NA reference DNA and MALBAC at high sequencing depth. The NA sample reached a genome coverage of 19.19-fold with 91.64% of the human genome (hg19) covered, the TruePrime-amplified sample had a genome coverage of 19.65-fold with a fractional coverage of 91.26% relating to an absolute difference in bases with 0 coverage of 11.7 million (Table 1). In comparison, the commercial RP-MDA method reached 85.57% genome coverage breadth at comparable read depth and MALBAC reached 58.57% coverage breadth. Nucleotide error rates in the reads were similar between the four samples (Table 1).

We also looked at coverage breadth saturation with increasing read input at a minimum coverage of 1 × (Fig. 5d, upper panel) and the deviation from the expected coverage using a Poisson distribution model (Fig. 5d, middle panel). The last panel in Fig. 5d shows the saturation of genome coverage breadth at a minimal coverage depth of 10 × . The TruePrime-amplified sample shows the highest similarity in all analyses to the NA material.

Relative coverage per chromosome is in general similar between the NA sample and TruePrime but with visible exception for some chromosomes that show a relatively lower coverage by TruePrime (for example, 19 and 22; Fig. 5e). We investigated whether this was related to the varying GC content between human chromosomes. The notable difference between chromosomal coverage in the NA sample is due to some other basic bias in the library prep or Illumina sequencing protocol, as the effect of GC content on chromosomal coverage does not reach significance ($P = 0.12$; Supplementary Fig. 5a). In the True-Prime-amplified sample there is a significant effect of GC content on chromosomal coverage ($R^2 = 0.38$; $P = 0.0017$; Supplementary Fig. 5b). Surprisingly, the behaviour of the commercial MDA (REPLI-g; Qiagen) is identical to this ($R^2 = 0.44$, $P = 0.0006$; Supplementary Fig. 5c,d), implying that the main driver behind this behaviour is Φ29DNApol, not the priming mechanism. A regression model using both chromosomal GC content and the variation of chromosomal coverage already present in the NA sample fully explains the pattern of chromosomal coverage in both amplified samples with an $R^2$ of 0.94 ($P < 0.0001$ for all effects), meaning that there is an unexplained variance of only 6% in this coverage behaviour. It is however important to note that the chromosomal variation inherent in the sequencing process is the most influential factor for the coverage pattern seen in the amplified samples.

In general, read number frequency in dependence of GC content appears similar between the NA and the TruePrime-amplified sample (Fig. 5f), with the exception of a slight preference of the TruePrime amplification reaction for a GC

range of 16–24%. MALBACs showed a right shift of the distribution curve.

We assessed coverage characteristics at a sequencing depth of ∼20× also by examining k-mer frequency distribution[48] (Fig. 6). K-mer frequencies were calculated using jellyfish[49].

K-mer size was set to 19 as suggested by Kelley *et al.*[50]. For the frequency plot shown the *y* axis was cut at 5% to show the higher frequencies at greater detail. The fraction of K-mers at a frequency of one (unique K-mers) was 34.36% for NA, 32.18% for TruePrime, 49.12% for MALBAC and 45.19% for the

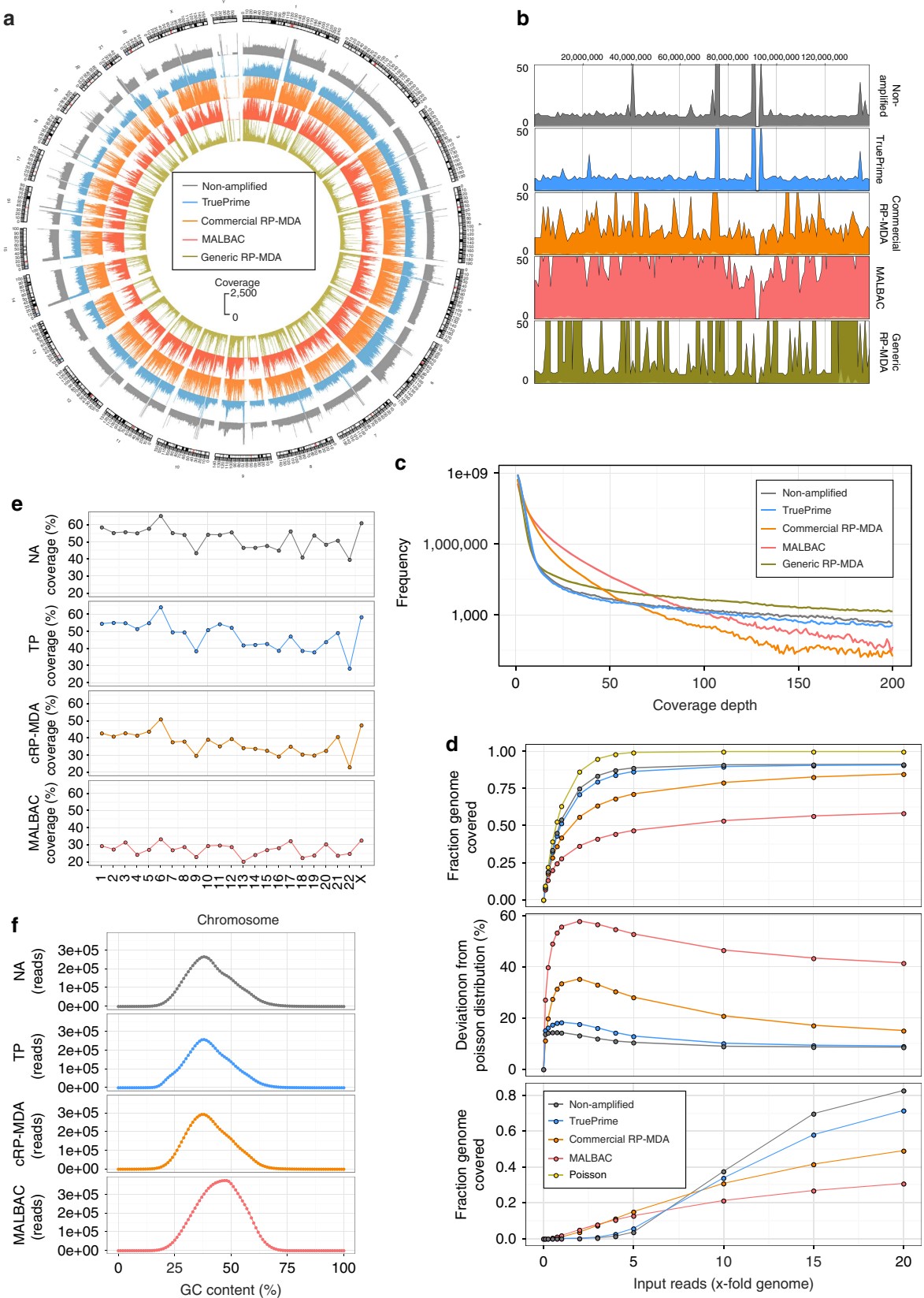

commercial RP-MDA protocol; the fraction of K-mers with frequencies of one or two was 36.32% for NA, 36.30% for TruePrime, 60.80% for MALBAC and 52.12% for the commercial RP-MDA protocol. Unique K-mer frequency is thought to be mostly due to nucleotide errors, but can also arise from a high fraction of very low coverage regions, which is possibly the explanation for the higher unique K-mer content in the commercial RP-MDA sample. On the other side, the identical unique K-mer content in the NA and TruePrime-amplified sample supports both the low nucleotide error rate of the amplification method and the even coverage obtained. The bimodal distribution of K-mer frequencies in the human genome is completely lost in the commercial RP-MDA and the MALBAC

sample, also most likely to be the effect of coverage inequality, whereas the distribution of the higher frequency k-mers is very similar between NA and TruePrime (peak depth 16 for NA, 13 for TruePrime).

Next, we looked at the reproducibility of the amplification results. Figure 7a shows a Circos plot of genome coverage from NA material (grey) and four single HEK293 cells amplified with TruePrime (blue) (input: exactly five million randomly selected read pairs). The fraction of the genome covered at this read number was 28% for the NA sample and between 26 and 28% for all 4 cells amplified. Cross-correlation between read numbers per 100 kb bin (Fig. 7b) and a sliding window view on chromosome 4 (Fig. 7c) highlight the similarity between the four replicates.

**Table 1 | Mapping and coverage parameters at high sequencing depth.**

|  | NA | TruePrime (1c) | Commercial RP-MDA | MALBAC |
|---|---|---|---|---|
| *Mapping characteristics* |  |  |  |  |
| Total read count | 509,313,411 | 536,789,206 | 560,053,809 | 527,319,977 |
| Total read length | 64,173,489,786 | 67,635,439,956 | 70,566,779,934 | 66,442,317,102 |
| Total reference length | 3,095,693,983 | 3,095,693,983 | 3,095,693,983 | 3,095,693,983 |
| Formal average coverage | 20.61 | 21.64 | 22.54 | 20.22 |
| Reads in aligned pairs | 471,412,468 | 482,799,006 | 504,010,950 | 478,462,898 |
| Reads in broken pairs: wrong distance or mate inverted | 11,768,592 | 31,849,758 | 31,794,142 | 18,026,124 |
| Duplicates (%) | 1.79 | 1.23 | 1.33 | 0.27 |
|  |  |  |  |  |
| *Zero coverage* |  |  |  |  |
| Count | 77,214 | 173,115 | 1,251,358 | 2,704,811 |
| Minimum length | 1 | 1 | 1 | 1 |
| Maximum length | 30,000,187 | 30,000,464 | 30,011,442 | 30,001,753 |
| Mean length | 3,351 | 1,562 | 357 | 470.79 |
| Standard deviation | 210,644 | 140,690 | 52,335 | 35,645.23 |
| Total length | 258,761,464 | 270,454,431 | 446,801,369 | 1,273,398,011 |
|  |  |  |  |  |
| Average coverage (true) *x*-fold | 19.19 | 19.65 | 20.51 | 19.47 |
| Genome fraction covered (%) | 91.64 | 91.26 | 85.57 | 58.87 |
|  |  |  |  |  |
| *Nucleotide errors (differing bases in %)* |  |  |  |  |
| A in ref | 0.63 | 0.87 | 0.45 | 0.81 |
| C in ref | 1.09 | 1.05 | 0.51 | 0.81 |
| G in ref | 0.83 | 0.94 | 0.51 | 0.81 |
| T in ref | 0.64 | 0.83 | 0.45 | 0.81 |
| - in ref | 0.73 | 0.26 | 3.03 | 2.89 |
|  |  |  |  |  |
| Total nucleotide errors | 0.77 | 0.83 | 0.50 | 0.85 |

MALBAC, multiple annealing and looping-based amplification cycle; MDA, multiple displacement amplification; NA, non-amplified.
The NA reference DNA and single HEK293 cells amplified with TruePrime, the commercial RP-MDA kit and MALBAC were sequenced at high depth, reaching a coverage of 19–20 × genome equivalents (mapped read pairs). The NA sample reached a genome coverage of 19.19 × with 91.64% of the human genome covered; the TruePrime-amplified sample showed a genome coverage of 19.65 × with a coverage of 91.26% (absolute difference in zero coverage of 11.7 million bases). The commercial RP-MDA sample reached 85.57% and the MALBAC sample reached 58.87% genome coverage.

**Figure 5 | Coverage comparison of PrimPol-mediated MDA of single cells to random primed MDA.** Comparison of genome coverage of NA HEK293 cell DNA and single human HEK293 cells amplified by either TruePrime, a generic RP-MDA, a commercial RP-MDA kit (REPLI-g single cell kit, Qiagen) or MALBAC. (**a**) Circos plot showing genome coverage using exactly 12 million read pairs as input. From outward to inward: NA-sample (grey), TruePrime (blue), commercial RP-MDA kit (orange), MALBAC (red) and generic RP-MDA protocol (yellow). Plot was generated using bin sizes of 50 kb. TruePrime shows a more even coverage distribution compared with the other methods. (**b**) Sliding window coverage comparison of chromosome 4 between NA (grey), TruePrime (blue), commercial RP-MDA kit (orange), MALBAC (red) and generic RP-MDA protocol (yellow). TruePrime shows a highly similar coverage pattern compared with NA. The commercial RP-MDA kit, MALBAC and generic RP-MDA display a higher jitter. The y axis is cut at a read depth of 50. For a detailed close-up, see Supplementary Fig. 4. (**c**) Comparison of coverage depth frequency distribution between NA sample and amplified samples (logarithmic y axis). The similarity of the TruePrime to the NA curves is noteworthy. (**d**) Coverage breadth saturation of human genome (fraction covered = y axis) at a minimum coverage of 1 × with increasing number of matched reads (x axis) (upper panel) and resulting deviation from expected coverage using a Poisson distribution model (middle panel). The lower panel shows the saturation curve for a minimal coverage depth of 10 ×. The TruePrime-amplified sample shows the highest similarity in all analyses to the NA material. (**e**) Coverage over different chromosomes (% chromosome covered) for NA (grey), TruePrime (blue), commercial RP-MDA kit (orange) and MALBAC (red). Coverage of NA, TruePrime and commercial RP-MDA kit are relatively similar, whereas MALBAC shows some differences and an overall lower coverage for some of the chromosomes. (**f**) Comparison of GC-content dependency of read frequency between NA (grey), TruePrime (blue), the commercial RP-MDA kit (orange) and MALBAC (red). Again, the behaviour between NA, TruePrime and commercial RP-MDA kit is very similar. MALBAC shows a right shift.

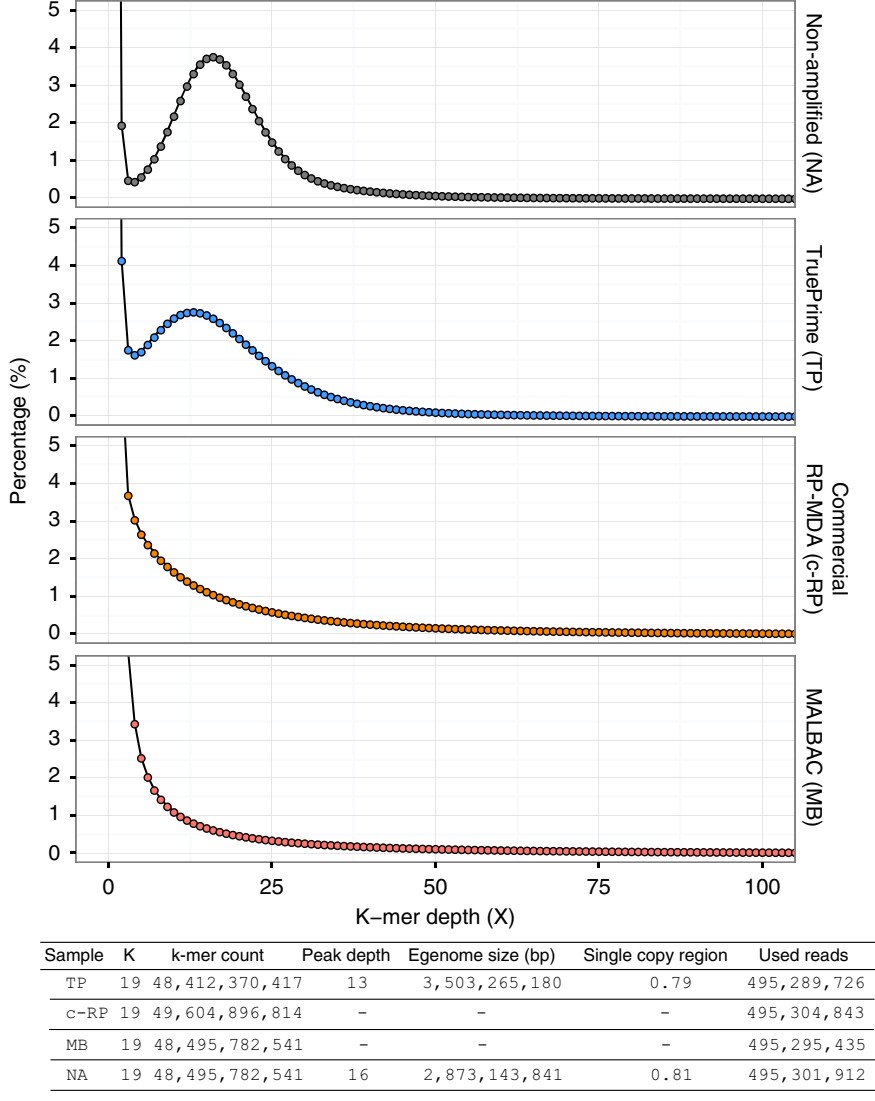

| Sample | K | k-mer count | Peak depth | Egenome size (bp) | Single copy region | Used reads |
|--------|---|-------------|------------|-------------------|--------------------|-----------|
| TP | 19 | 48,412,370,417 | 13 | 3,503,265,180 | 0.79 | 495,289,726 |
| c-RP | 19 | 49,604,896,814 | – | – | – | 495,304,843 |
| MB | 19 | 48,495,782,541 | – | – | – | 495,295,435 |
| NA | 19 | 48,495,782,541 | 16 | 2,873,143,841 | 0.81 | 495,301,912 |

**Figure 6 | K-mer comparison** K-mers were calculated with jellyfish. The $x$ axis shows the k-mer bin size, the $y$ axis shows the frequency of the corresponding k-mer bin in per cent. The NA sample and TruePrime show a clear separation of signal k-mer from error noise at low bin sizes, the commercial RP-MDA kit and MALBAC display no such separation. The estimated genome sizes $E$ were calculated on the basis of the k-mer distribution as $E = \sum \frac{B*F}{P}$, where $B$ is the bin size, $F$ is the corresponding frequency and $P$ is the peak depth estimated from the k-mer distribution. Only k-mer bins with size $>4$ were included, as lower bin sizes tend to be random noise originating from the library construction and sequencing process. Single-copy region contains the amount of k-mers covering the single copy peak from bin size 4 to bin size 50.

CNV detection in single cells is of particular interest in oncology. HEK293 cells have a partial aneuploidic state[51] and therefore CNV alterations should be detectable. Currently, a vast variety of bioinformatic tools are available for CNV detection based on different strategies[52,53]. We used both FREEC[54] and the recently published Ginkgo platform specifically optimized for single-cell CNV detection[55]. Visual comparison of CNV plots shows that TruePrime much better preserves the chromosomal CNV state than both RP-MDA and MALBAC (Supplementary Figs 6 (FreeC) and 7 (Ginkgo)). For Ginkgo, the median absolute deviation (MAD) of all pairwise differences in read counts between neighbouring bins[55] calculated for single HEK293 amplified with TruePrime was ∼0.2, a number close to the MAD value derived for published DOP-PCR amplified genomes, and strikingly different from the MADs from published RP-MDA methods that range between 0.35 and 0.8 (ref. 55).

Chimera formation is thought to be a problem in MDA potentially arising by strand switching during the displacement process[56]. We estimated the number of chimeras formed during the amplification process as the increase in broken read pairs due to wrong distance or mate inversion in the amplified samples relative to the NA sample. Although the NA sample had 2.5% broken read pairs of this nature, the TruePrime-amplified samples showed between 3.9 and 6.5%, suggesting that there is an increase in chimeras generated by the TruePrime process in the range of 2–3% over all read pairs. The commercial RP-MDA protocol showed a similar fraction of broken read pairs due to wrong distance or mate inversion (5%) as the TruePrime sample implicating the same increase in chimeras and suggests that the priming process has little influence on the occurrence of chimeras in Φ29DNApol-mediated DNA amplification protocols.

For SNV calling, we used four different SNV callers due to high inter-caller variability[57,58] (Supplementary Table 1). With the exception of samtools/bcftools, all callers detected similar numbers of SNVs with a median of 3.0 Mio SNVs for the NA and 2.7 Mio SNVs for the TruePrime-amplified cell. The overlap

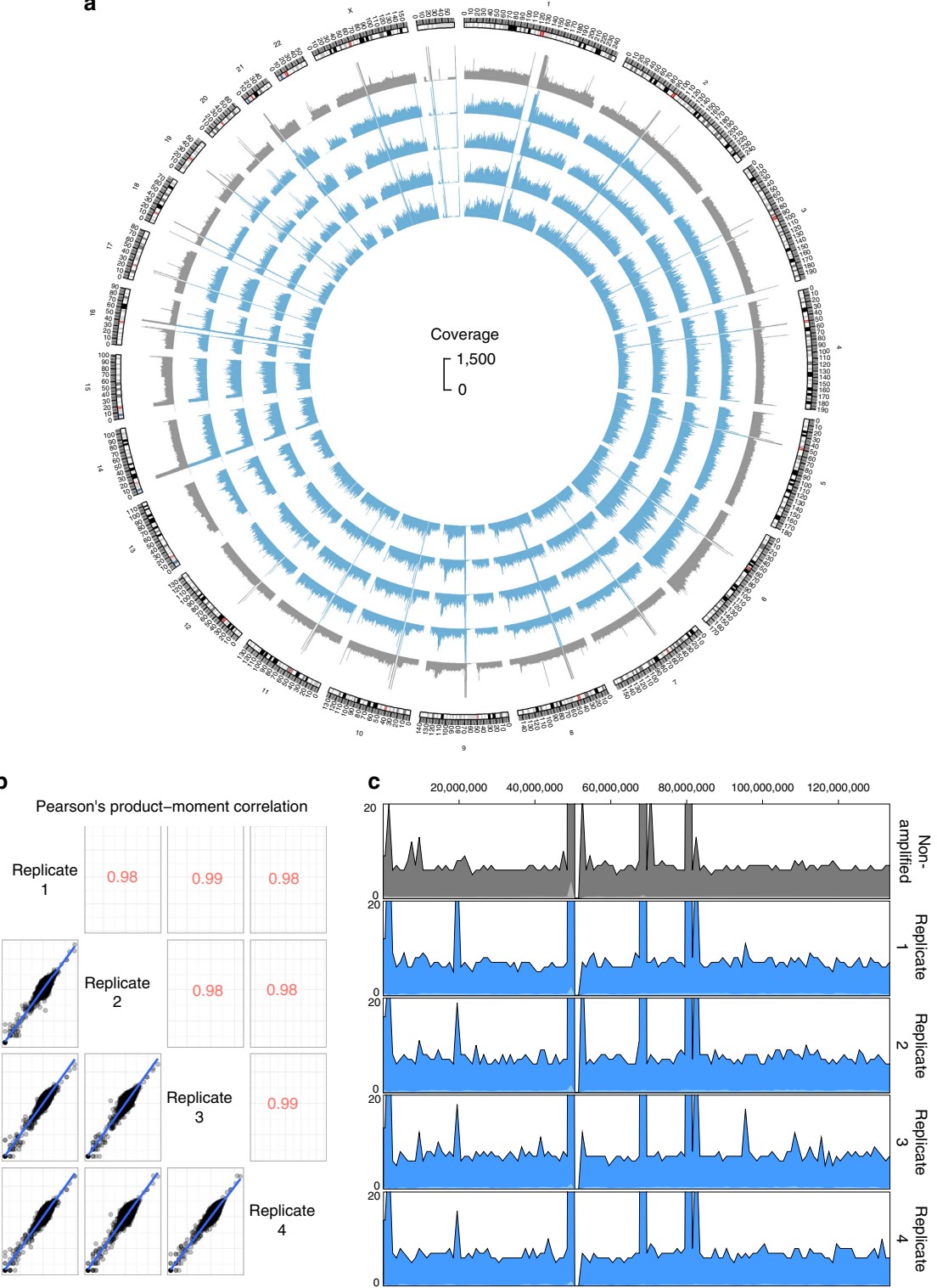

**Figure 7 | Reproducibility in TruePrime-amplified single cells.** (**a**) Circos plot of genome coverage from NA material (grey) and four HEK293 cells amplified with TruePrime (blue) (input: exactly five million randomly selected read pairs). The coverage pattern appears very similar in the 4 replicates. It is noteworthy that no major part of any chromosome is missing in the amplified DNA. (**b**) Pearson's product–moment cross-correlation of binned read depth (bin size = 100 kb) for all four replicates. (**c**) Sliding window coverage comparison of chromosome 4 between NA and the four replicates. Again, the coverage pattern is highly similar between the replicates. For a close-up view, see Supplementary Fig. 8.

between SNVs in the two samples was 2.4 Mio, equivalent to 81% of the SNVs found in the NA sample (Supplementary Table 1). In contrast, in the cell amplified by the commercial RP-MDA method, only 1.6 Mio SNVs were detected of which 1.4 Mio

overlapped with SNVs detected in the NA sample (45% of all NA SNVs; Supplementary Table 1). This was even lower in MALBAC, where only 30% of the SNVs detected in the NA sample were recovered.

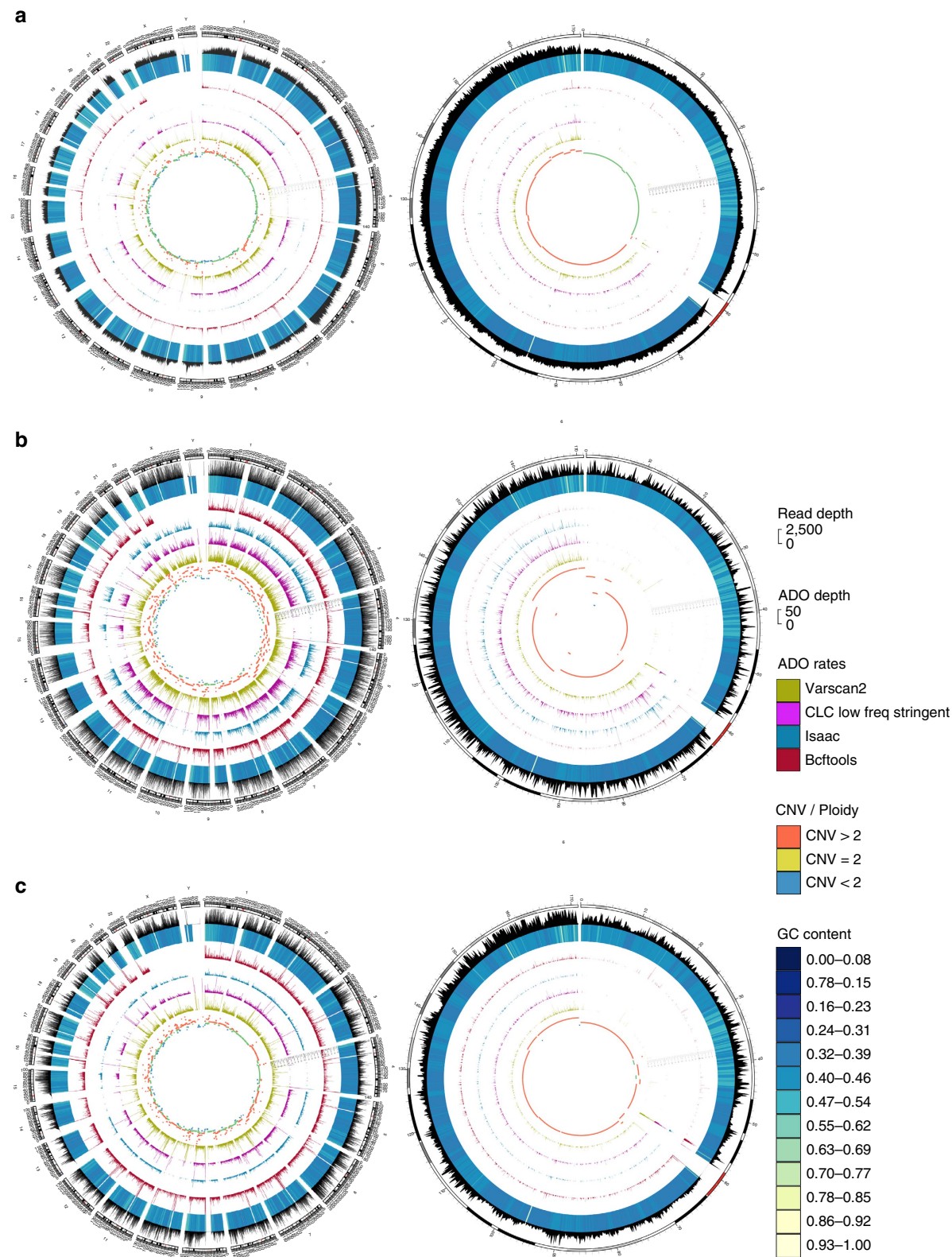

**Figure 8 | ADOs and CNVs.** Circos plots showing heterozygote–homozygote conversion events (labelled as 'ADO rates') collected from different SNV callers. From outward to inward: overall coverage as shown in Fig. 5 (bin size 50 kb, y axis from 0 to 2,500 reads), GC content, heterozygote–homozygote conversion events calculated from samtools/bcftools, Isaac variant caller, CLC low frequency caller, varscan2 (bin size 100 kb, y axis range from 0 to 50) and calculated ploidy state/CNVs using FreeC (bin size of 50 kb). The left Circos plot shows a genome-wide representation, the right Circos plot depicts chromosome 6 only for (**a**) TruePrime versus none-amplified, (**b**) commercial RP-MDA kit versus NA and (**c**) MALBAC versus NA. Coverage and calculated ploidy state reflect the amplified sample in each plot. TruePrime has a much lower number of ADOs than both the commercial RP-MDA kit and MALBAC. Likewise, the CNVs detected in the commercial RP-MDA kit and MALBAC-amplified samples deviate clearly from the TruePrime sample. For details regarding the CNVs in detail, see Supplementary Figs 6 and 7.

A major question with WGA methods is the so-called ADO rate, meaning the fraction of heterozygous SNVs that are lost due to exclusive or predominant amplification of only one allele. A way to estimate this number is to establish the heterozygous SNVs in the NA sample and determine the fraction that is called as homozygous SNVs in the amplified sample. The overall number of SNVs that were detected as heterozygous in the NA sample and homozygous in the TruePrime-amplified sample (1c) ranged from 0.73% (Isaac SNV caller) to 8.42% (Varscan2) with a median of 5.95% across the whole genome (Supplementary Table 1), suggesting an ADO of ∼1.45–15.5% with a median of 11.23% (AB->AA plus the non-observed AB->BB). Interestingly, there was a considerable degree of variation in the apparent conversion rate among different chromosomes (Fig. 8). In contrast, the commercial RP-MDA method showed a very high estimated ADO rate of 45.74% (median of the 4 callers) similar to the MALBAC method (47.22% in the median; Supplementary Table 1).

Related to the question of ADOs is the issue of false positives that could be generated during the amplification process. The TruePrime false positive rate (FPR) for the SNVs based on the overlap with the NA sample was around 1% for three of the callers and 3.66% for samtools/bcftools (Supplementary Table 1). The RP-MDA FPR was similar to that (Supplementary Table 1). MALBACs showed the highest FPR with 5.9%. Another possibility to detect generation of false positives is to determine the conversion of homozygote alleles to heterozygotes for a haploid chromosome (#18) in our HEK293 cell line. This ensures that the homozygote calls from the NA DNA are true positives. The rates here were 0.13–1.29% for TruePrime, depending on the caller, and similar in the RP-MDA sample (Supplementary Table 1).

## Discussion

Here we have cloned, characterized and put into technical use a novel PrimPol, TthPrimPol. To our knowledge, this is the first instance that this class of enzymes has been made available for biotechnological applications.

We have exploited several unique features of TthPrimPol for its cooperation with Φ29DNApol, to enable successful WGA. First, the ability to use dNTPs and synthesize DNA primers makes it possible for the enzyme to work without addition of NTPs to the reaction, which would alter DNA polymerase characteristics of Φ29DNApol and would generate RNA/DNA chimeric molecules with possible disadvantageous consequences for downstream enzymatic manipulation for library construction and so on. Second, TthPrimPol works with $Mg^{2+}$ as the only metal ion and does not need $Mn^{2+}$, which would interfere with the high-fidelity DNA synthesis by Φ29DNApol. Third, the primase function of TthPrimPol synthesizes DNA primers of 7–9 nucleotides length in a processive mode, but then switches to a distributive mode for its polymerase function, enabling the highly processive Φ29DNApol to take over elongation of those primers. This unique compatibility of the two enzymes thus enables the replacement of the error-prone polymerase function of TthPrimPol with the high-fidelity Φ29DNApol for WGA (Fig. 9).

We find that TruePrime has an exquisite breadth of coverage, which approximates that of NA DNA (91.26% at ∼19× coverage). Breadth of coverage is a known strength of MDA-based protocols including the hybrid MALBAC method[5,21,59] as opposed to purely PCR-based methods (DOP-PCR has only ∼10% coverage breadth[3,60]) and reaches ranges of over 90% genome coverage[61]. In our hands, the commercial RP-MDA protocol gave a coverage breadth of ∼86%, whereas MALBAC reached 59%. Together with the inherently high fidelity of Φ29DNApol of about $10^{-7}$ (ref. 18) and the high evenness of coverage, this lays ideal foundations for high-quality SNV calling throughout the genome of single cells. Indeed, we report an 80.6% concordance of SNVs called in NA and amplified samples in the range of expected SNV numbers. Our estimate for the ADO number appears low (estimated at 11.23% in the median of four

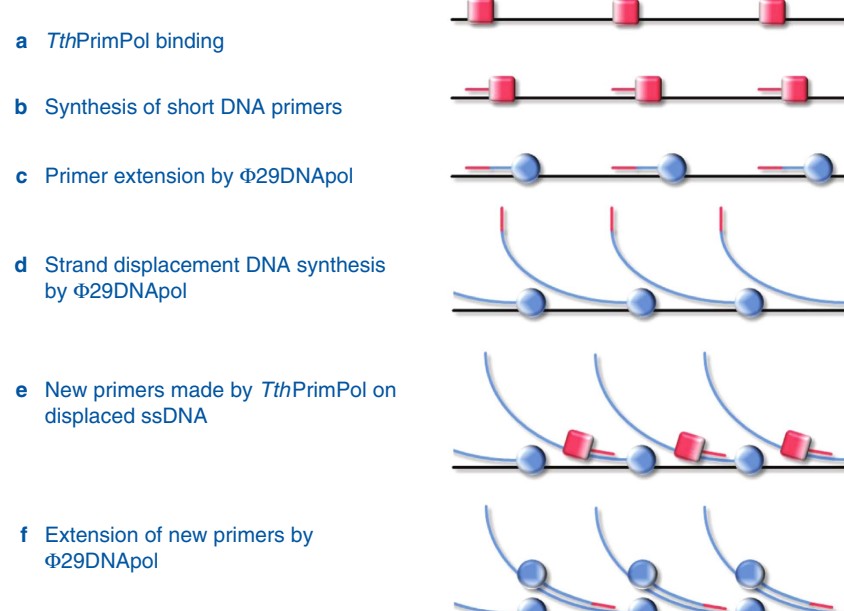

**a** TthPrimPol binding

**b** Synthesis of short DNA primers

**c** Primer extension by Φ29DNApol

**d** Strand displacement DNA synthesis by Φ29DNApol

**e** New primers made by TthPrimPol on displaced ssDNA

**f** Extension of new primers by Φ29DNApol

**Figure 9 | Scheme of the TruePrime reaction.** TruePrime reaction steps leading to amplification of (genomic) DNA. (**a**) TthPrimPol binds to denatured DNA at different sites. (**b**) TthPrimPol synthesizes short DNA primers. (**c**) DNA primers are recognized by Φ29DNApol, which replaces TthPrimPol extending the primers. (**d**) DNA primer elongation coupled to strand displacement by Φ29DNApol leads to the exposure of new single-stranded template regions. (**e**) TthPrimPol catalyses new rounds of priming on the displaced ssDNA. (**f**) New DNA primers trigger further rounds of strand-displacement synthesis, leading to exponential amplification of the target DNA.

SNV callers and as low as 1.45% in the Isaac caller). Numbers in the literature for MDA methods for single-genome cell amplification vary widely between 4 and 50% (refs 62–64), and we find a high estimated ADO for both RP-MDA and MALBAC ($>45\%$). It is important to note that estimation of the ADO using the heterozygote to homozygote conversion is also subject to the variant caller used and the read depth at each SNV locus. We attempted to address this issue by using four different callers and reporting detailed output parameters. The much higher ADO in the RP-MDA and MALBAC sample may be partially due to the lower evenness in coverage, which results in more loci having lower coverage and therefore having a higher chance of missing one allele.

A weakness of the MDA group of methods despite the superior breadth of genome coverage is CNV detection, in particular for methods relying on read-depth counting. This is due to inequality bias in amplifying different genomic regions. TruePrime has considerably less amplification bias than random primer-based protocols and reaches the coverage dispersion characteristics of DOP-PCR amplification experiments from single cells. Consequently, this allows for improved CNV detection accuracy, thus improving one major weakness of Φ29DNApol/MDA-based protocols so far.

Another reported problem with MDA methods concerns chimera formation occurring by strand switching during strand displacement. We find that percentage of broken read pairs probably due to chimeras is at a 2–3% percentage fraction of reads. In summary, TruePrime presents an important improvement to Φ29DNApol-mediated amplification of single-cell genomes. We believe that this method will contribute greatly to the accessibility of genomic information from single cells.

## Methods

**Computational modelling of *Tth*PrimPol 3D structure.** The 3D structure of *Tth*PrimPol was modelled using as template the crystallographic structure of the DNA primase/polymerase domain of ORF904 from the archaeal plasmid pRN1 (PDB ID:3M1M and 1RNI). *Tth*PrimPol amino acid residues 4 to 166 were modelled with the Phyre2 (ref. 59) online server using 3M1M as template; *Tth*PrimPol amino acid residues 167 to 208 were modelled with the DeepView Project Mode of Swiss Model[65] online server using 1RNI as template. DNA template and primer strands, metals and incoming nucleotide were modelled using two crystal structures of the polymerase domain (PolDom) of *M. tuberculosis* ligase D (PDB ID:4MKY for template/primer and PDB ID:3PKY, for metals and incoming nucleotide), which were fitted to the *Tth*PrimPol model by using the three invariant catalytic aspartates (motifs A and C) and the invariant histidine (at motif B) as reference coordinates. The image depicted in Fig. 1b was created with the PyMol Molecular Graphics System (version 1.2r3pre, Schrödinger, LLC), omitting the amino acid sequence information from the LigD PolDom crystals.

**Cloning of *Tth*PrimPol.** Sequence analysis of the *T. thermophilus* HB27 genome (DDBJ/EMBL/GeneBank AE017221.1; GI:46197919) revealed the ORF TTC0656, encoding a protein that belongs to the AEP superfamily. Using this sequence information, we synthesized two primers (5′-ccggcccatatgaggccgattgagcacgccc-3′ and 5′-gcgcgcgaattctcatacccacctcctcatccggg-3′) for amplification of the *Tth*PrimPol gene by PCR from *T. thermophilus* genomic DNA. The gene fragment amplified by PCR using Expand High Fidelity polymerase (Roche Diagnostics, Mannheim, Germany) was ligated into the pGEM T-easy vector (Promega, Madison, WI, USA) by TA cloning and confirmed by sequencing. Using the NdeI and EcoRI sites, the fragment bearing the target gene was ligated into pET28 vector (Novagen, Merck-Millipore, Billerica, MA, USA), allowing the expression of *Tth*PrimPol fused with a multifunctional leader peptide containing a hexahistidyl sequence for purification on $Ni^{2+}$-affinity resins.

**TthPrimPol production.** Expression of *Tth*PrimPol was carried out in the *E. coli* strain BL21-CodonPlus (DE3)-RIL (Stratagene), with extra copies of the argU, ileY and leuW transfer RNA genes. Expression of *Tth*PrimPol was induced by the addition of 1 mM isopropyl-β-D-thiogalactoside to 1.5 l of log phase *E. coli* cells grown at 30 °C in lysogeny broth (LB) to an Abs$_{600\,nm}$ of 0.5. After induction, cells were incubated at 30 °C for 5 h. Subsequently, the cultured cells were harvested and the pelleted cells were weighed and frozen ($-20\,°C$). Just before purification, which was carried out at 4 °C, frozen cells (3.5 g) were thawed and resuspended in 20 ml buffer A (50 mM Tris-HCl pH 7.5, 5% glycerol, 0.5 mM EDTA and 1 mM

dithiothreitol (DTT)) supplemented with 1 M NaCl, 0.25% Tween-20 and 30 mM imidazole, and then disrupted by sonication on ice. Cell debris and insoluble material were discarded after a 50 min centrifugation at 40,000 g. The supernatant was loaded into a HisTrap crude FF column (5 ml, GE Healthcare) equilibrated previously in buffer A supplemented with 1 M NaCl, 0.25% Tween-20 and 30 mM imidazole. After exhaustive washing with buffer A supplemented with 1 M NaCl, 0.25% Tween-20 and 30 mM imidazole, proteins were eluted with a linear gradient of 30–250 mM imidazole. The eluate containing *Tth*PrimPol was diluted with buffer A supplemented with 0.25% Tween-20 to a final 0.1 M NaCl concentration and loaded into a HiTrap Heparin HP column (5 ml, GE Healthcare), equilibrated previously in buffer A supplemented with 0.1 M NaCl and 0.25% Tween-20. The column was washed and the protein eluted with buffer A supplemented with 1 M NaCl and 0.25% Tween-20. This fraction contains highly purified ($>99\%$) *Tth*PrimPol. Protein concentration was estimated by densitometry of Coomassie Blue-stained 10% SDS–polyacrylamide gels, using standards of known concentration. The final fraction, adjusted to 50% (v/v) glycerol, was stored at $-80\,°C$.

**Primase assays.** 3′-GTCC-5′ oligonucleotide (1 μM) or its variant XTCC oligonucleotides (1 μM) or M13mp18 single-stranded DNA (ssDNA) (20 ng μl$^{-1}$) were used as alternative templates to assay primase activity. The reaction mixtures (20 μl) contained 50 mM Tris-HCl pH 7.5, 75 mM NaCl, 5 mM $MgCl_2$ or 1 mM $MnCl_2$, 1 mM DTT, 2.5% glycerol, 0.1 mg ml$^{-1}$ BSA, [α-$^{32}$P] dATP (16 nM; 3,000 Ci mmol$^{-1}$) or [γ-$^{32}$P] ATP (16 nM; 3,000 Ci mmol$^{-1}$), the indicated amounts of each dNTP or NTP, in the presence of *Tth*PrimPol (400 nM). After 60 min at either 55 °C or 30 °C, as indicated, reactions were stopped by addition of formamide loading buffer (10 mM EDTA, 95% v/v formamide and 0.3% w/v xylene cyanol). Reactions were loaded in 8 M urea-containing 20% polyacrylamide sequencing gels. After electrophoresis, de novo synthesized polynucleotides (primers) were detected by autoradiography.

To evaluate the processivity of primer synthesis by *Tth*PrimPol, we used heparin as a competitor (Fig. 3c). *Tth*PrimPol (10 nM) was pre-incubated for 5 min on ice in the previously described reaction buffer, either in the absence/presence of heparin (1 ng μl$^{-1}$). Subsequently, the reaction was complemented with M13mp18 ssDNA (5 ng μl$^{-1}$), dATP, dCTP and dTTP (10 μM each), [α-$^{32}$P] dGTP (16 nM; 3,000 Ci mmol$^{-1}$) and heparin (1 ng μl$^{-1}$) when indicated, and the incubation was maintained for 10 min at 30 °C and processed as described.

**Primase/polymerase-coupled assay.** A 'Pulse and Chase' experiment was designed to analyse the extension by Φ29DNApol of the primers synthesized by *Tth*PrimPol in two consecutive stages (pulse and chase), as indicated in Fig. 3d. During pulse, the reaction mixtures (20 μl; 50 mM Tris-HCl pH 7.5, 75 mM NaCl, 10 mM $MgCl_2$, 1 mM DTT, 2.5% glycerol and 0.1 mg ml$^{-1}$ BSA) containing decreasing concentrations of *Tth*PrimPol (100, 25 and 6.25 nM), [α-$^{32}$P] dGTP (16 nM), dATP + dCTP + dTTP (1 μM) and 5 ng μl$^{-1}$ M13mp18 ssDNA were incubated at 30 °C during 20 min. Half of the reaction was analysed as described for the primase assays. During chase, a second half of the reaction was supplemented with Φ29DNApol (50 nM) and the four unlabelled dNTPs (10 μM), to allow primer extension for another 20 min at 30 °C. Then, the samples were processed as described.

**Rolling circle amplification.** M13mp18 circular ssDNA was used as input for the TruePrime RCA kit workflow. Briefly, DNA (2.5 μl; 40 fg μl$^{-1}$) was first denatured by adding 2.5 μl of alkaline buffer D and incubated 3 min at room temperature. The samples were then neutralized by adding 2.5 μl of buffer N. The amplification mix containing 9.3 μl of H$_2$O, 2.5 μl of reaction buffer, 2.5 μl of dNTPs, 2.5 μl of Enzyme 1 (*Tth*PrimPol) and 0.7 μl of Enzyme 2 (Φ29DNApol) was added to the DNA samples, resulting in a final reaction volume of 25 μl. When indicated, *Tth*PrimPol was replaced by *Hs*PrimPol or random synthetic primers (50 μM). Reaction mixtures were incubated for 3 h at 30 °C and Φ29DNApol was inactivated for 10 min at 65 °C, to avoid degradation of the amplification products. Amplified DNA was quantified using the Quant-iT PicoGreen dsDNA Assay Kit (Invitrogen, Life Technologies, Carlsbad, CA, USA) following the recommendations of the manufacturer. Briefly, samples were diluted 1:1,000 in 1× TE and, in parallel, a DNA standard using human genomic DNA (Roche) with 1.6, 0.8, 0.4, 0.2 and 0.1 μg ml$^{-1}$ was prepared. Twenty microlitres of the sample or DNA standard were transferred into a 96-well plate and 20 μl of PicoGreen working solution (PicoGreen stock solution 1:150 diluted) was added. After gently shaking the 96-well plate, fluorescence was measured in a Fluostar Microplate Reader (BMG Labtech; excitation: 485 nm and emission: 520 nm). For measurements, duplicates for each sample and DNA standard were performed, and DNA concentration was determined from the human genomic DNA standard curve.

**Whole genome amplification.** Six picograms (Fig. 4, part b) or different doses ranging from 1 ng to 100 ag (Fig. 4, part d) of human genomic DNA (Promega) were used as input in the reactions. Input DNA was subjected to the TruePrime WGA kit workflow. Briefly, DNA (2.5 μl) was first denatured by adding 2.5 μl of buffer D and incubating 3 min at room temperature. The samples were then neutralized by adding 2.5 μl of buffer N. The amplification mix containing 26.8 μl of H$_2$O, 5 μl of reaction buffer, 5 μl of dNTPs, 5 μl of Enzyme 1 (*Tth*PrimPol) and

0.7 µl of Enzyme 2 (Φ29DNApol) was added to the DNA samples, resulting in a final reaction volume of 50 µl. When indicated, *Tth*PrimPol was replaced by the same concentration of *Hs*PrimPol or random synthetic primers (50 µM). Reaction mixtures were incubated for 3 h at 30 °C and Φ29DNApol was inactivated for 10 min at 65 °C, to avoid degradation of the amplification products. Amplified DNA was quantified using the Quant-iT PicoGreen dsDNA Assay Kit (Invitrogen, Life Technologies).

**Single-cell WGA.** We chose HEK293 cells for testing WGA protocols due to their partial aneuploidic state[51]. Remark: HEK293 cells are listed as potentially contaminated with HeLa cells in ICLAC (http://iclac.org/wp-content/uploads/Cross-Contaminations-v7_2.pdf) based on a publication in 1981. The cells analysed by us are clearly HEK293 cells based on their genomic sequence and CNV profile. The cell line has been obtained in 2010 from the DSMZ (Leibniz-Institute German Collection of Microorganisms and Cell Cultures; DSMZ number: ACC305). The cell line is regularly checked for mycoplasma contamination by a PCR assay (primer A: 5′-ggc gaa tgg gtg agt aac acg-3′ and primer B: 5′-cgg ata acg ctt gcg acc tat-3′).

HEK293 cells were washed with 1 × PBS, followed by incubation with Trypsin-EDTA solution (Gibco). After resuspending cells with culture medium, they were spun down and washed again with 1 × PBS. After preparing three serial dilutions of cells in 1 × PBS, they were counted, diluted to a final concentration of 1 cell per 2.5 µl 1 × PBS. This volume was dispensed into clear-well plates and visually inspected. TruePrime Single Cell WGA kit workflow was followed to amplify the genomic DNA of each cell. Briefly, 2.5 µl of lysis buffer L2 were added, followed by incubation for 10 min on ice. To neutralize the lysis buffer, 2.5 µl of neutralization buffer N were added. The amplification mix containing 26.8 µl of H$_2$O, 5 µl of reaction buffer, 5 µl of dNTPs, 5 µl of Enzyme 1 (*Tth*PrimPol) and 0.7 µl of Enzyme 2 (Φ29DNApol) was added to the neutralized samples, resulting in a final reaction volume of 50 µl. The reaction mixtures were incubated for 3 or 6 h at 30 °C, followed by inactivation of Φ29DNApol for 10 min at 65 °C. To obtain NA reference DNA, genomic DNA was extracted from HEK293 cells using QIAamp genomic DNA extraction kits (Qiagen). DNA concentration was determined by the Quant-iT PicoGreen dsDNA Assay Kit (Invitrogen, Life Technologies). For amplification with the REPLI-g single-cell kit (Qiagen; commercial random primed MDA) and with the MALBAC protocol Single Cell WGA kit (Yikon Genomics), instructions of the manufacturers were followed.

**Sequencing.** After amplification, DNA was precipitated by ethanol precipitation. DNA fragmentation (Covaris), library preparation using NebNext (NEB) and paired end sequencing (HiSeq 2500, v4 chemistry, HiSeq Control Software, Version 2.2.58, Real-Time Analysis, Version 1.18.64 and Sequence Analysis Viewer, Version 1.8.46, Casava Version 1.8.2) were performed at GATC Biotech, Konstanz, Germany. FastQ files were obtained and further processed.

**Bioinformatic and statistical analyses.** *Quality assessment and mapping*. CLC Genomic Workbench Version 8.5 (Qiagen) was used for main analyses of NGS data sets (alignment and mapping parameters). Illumina BaseSpace FASTQC v1.0.0 was used for sequencing quality assessment and GC content dependency calculations. Circos plots were generated using the Circos framework[66].

*Saturation of coverage breadth*. For each 20 × sample, a calculated number of aligned reads were selected with samtools view function from bam files to produce the desired read depth (0.1 ×, 0.25 ×, 0.5 ×, 0.75 ×, 1 ×, 2 ×, 3 ×, 4 ×, 5 ×, 10 × and 15 ×). The fraction of single read coverage and the fraction of tenfold coverage was calculated with bedtools genomeCoverageBed function.

**SNV calling and analyses.** All SNV callers were applied to the same aligned sequence data set (BAM files aligned to hg19 with an overall coverage (mapped reads) of ~19–20 × of the human genome).

CLC Genomics Workbench 9.0 low-frequency variant detection caller[67] was used with the following stringent settings (required significance (%) = 1.0, ignore positions with coverage above = 1,000, restrict calling to target regions = not set, ignore broken pairs = yes, ignore nonspecific matches = reads, minimum coverage = 10, minimum count = 3, minimum frequency (%) = 5.0, base quality filter = yes, neighbourhood radius = 5, minimum central quality = 20, minimum neighbourhood quality = 15, read direction filter = yes, direction frequency (%) = 10.0, relative read direction filter = yes, significance (%) = 1.0, read position filter = yes, significance (%) = 1.0, remove pyro-error variants = no).

*Samtools 1.3*[68]—*mpileup/bcftools*, htslib 1.3.1 was used with -E and -uf settings. Bcftools 1.3.1 was used with -cv, -Ov and --ploidy = GRCh37.

VarScan2.v2.4.1[69] was used with standard settings.

Isaac Variant Caller 1.0.7[70] was used with the (standard) settings: isSkipDepthFilters = 0, maxInputDepth = 10,000, depthFilterMultiple = 3.0, indelMaxRefRepeat = −1, minMapq = 20, minGQX = 30, isWriteRealignedBam = 0, binSize = 25000000CLC. The Isaac variant caller uses the GATK Unified Genotyper followed by filtering with the variant quality score recalibration (VQSR) protocol[71].

SNV intersection analyses were done with Illumina BaseSpace VCAT (Variant Calling Assessment Tool v2.3.0.0) and Illumina VariantStudio v2.2.4

(https://basespace.illumina.com/home/index). Estimated ADO rate was determined over the whole genome using R v3.3.0 by determining the number of variants in the overlapping set of SNVs that were heterozygote in the NA sample and homozygote in the amplified samples and assuming an equal number of heterozygote to homozygote conversions that were not contained in the overlap set because of reversion to the reference allele.

**SNV false positive rates.** For the estimation of the FPR for each SNV caller all called SNVs from the NA sample were assumed as true SNVs. Thus:

$$TP = sample \cap control$$

$$FP = Number\ of\ SNVs\ in\ sample - TP$$

$$FN = Number\ of\ SNVs\ in - TP$$

$$TN = Number\ of\ positions\ in\ human\ genome - (TP + FP)$$

$$FPR = \frac{FP}{FP + TN}$$

**Position based recall and precision (calculated by V-CAT).** Recall and precision were calculated by the VCF Gold Standard Comparison with NIST Genome in a Bottle integrated calls v0.2.

$$Recall = \frac{TP}{TP + FN} \qquad Precision = \frac{TP}{TP + FP}$$

**CNV calling.** CNV analyses were done using Ginkgo[55] (http://qb.cshl.edu/ginkgo/) or ControlFreeC[54] (http://bioinfo-out.curie.fr/projects/freec/).

**K-mer frequency analyses.** K-mers were calculated with jellyfish[49]. K-mer size was set to 1,950. An estimation of genome sizes from k-mer counts were calculated by the bin sizes B, the corresponding frequency F and the peak depth estimated from the k-mer distribution: $E = \frac{\sum B*F}{P}$. The single-copy region was extracted visually as the first peak after the initial error peak at bin size 1.

**Statistical analyses.** Additional statistical analyses were done using JMP 12 (SAS Institute, Heidelberg, Germany) or R v3.3.0 /Rstudio v0.99.902. A *P*-value < 0.05 was considered significant.

**Data availability.** The data sets generated during and analysed during the current study are available in the GenBank Sequence Read Archive, accession number SRP085855.

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

## Acknowledgements

We thank Patricia Garrido, Frank Herzog, Gisela Eisenhardt, Patricia Rebollo and Clara López for expert technical help. We thank Margarita Salas for intense discussions. L.B. was funded by the Spanish Ministry of Economy and Competitiveness (BFU2012–37969 and CSD2007–00015), and by Comunidad de Madrid (S2011/BMD-2361). S.G.-G. was the recipient of a fellowship from the Spanish Ministry of Economy and Competitiveness.

## Author contributions

A.J.P., L.B. and A.S. conceived and directed the project and designed the experiments. A.J.P., C.K., D.W., S.G.-G., M.I.M.-J. and A.D.-T. performed the experiments. B.B., O.W. and A.S. conducted bioinformatic and statistical analyses. A.J.P., A.S. and L.B. wrote the manuscript. All authors have read and approved the final version of the manuscript.

## Additional information

**Competing financial interests:** A.J.P. and L.B. are inventors on patent applications incorporating part of the reported findings. A.J.P., B.B., D.W., O.W., A.S. and C.K. are or were employees of SYGNIS Biotech SLU/SYGNIS Bioscience GmbH & Co KG. L.B. owns shares of SYGNIS A.G. and is a paid advisor to the company. TruePrime is a commercial product line available through SYGNIS or its distributors. The remaining authors declare no competing financial interests.

