## [Peer Review File · Nature Communications]

Reviewers' comments:

Reviewer #1 (Remarks to the Author):

Picher et al describe a novel whole genome amplification method using TthPrimPol and Phi29. The results presented in the manuscript showed great promise for future single-cell genomic analysis. However, the manuscript was written like a product brochure instead of a scientific publication. Some of the my specific comments are as below:

1. In general, the manuscript releases scarce amount of technical details about the TruePrime WGA method and the following analyses. For example, the amplification was first carried out in two separate steps, with different amount of dNTP (Fig 2C). Then for the single cell section, a single step was employed. I understand that this is probably for the protection of the commercial product. However, as a scientific publication, the authors should try their best to facilitate the replication of their results. Given that previous MDA and MALBAC papers all described very detailed protocol, I suggest the authors follow the academic traditions.
2. The single-cell section needs some improvement. First, please list the details about your competitor R control. Again, this is not a product brochure. Second, please add some more comparisons. It is now well known that MDA gets the amplification bias from excess level of amplification. When amplification is limited to ~1000 folds, the bias is greatly improved. Following this direction, many studies have shown great results using certain kinds of microfluidics. Since we do not need micrograms of DNA for single-cell sequencing, and MDA can do better when getting just nano grams of DNA, it is not the best comparison I can expect for Fig 3 and 4. A direct comparison with MALBAC is necessary. 70% coverage stated on p8 is not what they claimed in the original publication.
3. It is not clear to me why the expected coverages in Fig 4A are different across different reactions, given that exact 12M reads were used. If the read quality was the case, the authors should describe.
4. Fig 4C should have Y-axis labels. It is not even clear whether it is linear or log. Same for Fig 5A.
5. Lower coverage on chr19 and 22 suggests GC effect on priming. chr19 and 22 have the highest GC contents across all human chromosomes. Actually the coverages were also lower for other chromosomes with high GC content. If this effect is true, I would speculate similar bias within each chromosome, such a promotor regions. Since the authors did not release finer alignment results, it is difficult to tell for now.
6. What is the bin size for each dot in Fig 5B.
7. Regarding the chimera formation, as the authors stated, it's from the strand switching during displacement process. The authors didn't show data from MDA, and I don't expect improvement over MDA by changing the priming mechanism. Whether 2-3% increase is a big number should be left for readers' judgement.
8. What is the definition of novel SNP on P7? If the authors think high false positive rate was due to software sensitivity, they should adjust the parameters. 25% false positive SNV calling rate is disturbing. 19x sequencing coverage is generally not enough for SNV calling. If the authors can not confidently identify SNV from the bulk sample, I can not trust the analyses on p7. The authors should describe the details about ADO calculation, i.e. which loci were used from the bulk and the single-cell samples.

In general, the technology presented in the manuscript is very promising and worth following. However, the manuscript was well written to demonstrate the technology.

Reviewer #2 (Remarks to the Author):

In this manuscript, the authors have cloned and characterized the *Thermus thermophilus* PrimPol enzyme and further created a novel primer-free WGA method based on this enzyme for single cell genome amplification. Overall, the manuscript is well written and the conclusions drawn from the data are appropriate. However, a few specific and important points must be addressed.

Specific points:

1. A seq ladder should be included for all PrimPol assays to deduce the exact lengths of the products.
2. Fig 2A and B, to further support the wide substrate specificity, the substrates with different sequence from the one used in 2A should be tested.
3. Fig 2C, the two gels of PULSE and CHASE are not well aligned and labeled. Do the highly elongated primers stay in the loading wells? why there are also small amounts of such long products in the PULSE stage (left gel)? Why there are more TthPrimPol-created primers (7-9base) left unextended in the decreased amounts of TthPrimPol?
4. For the processivity, a more stringent assay (e.g., heparin-based) should be used to ensure single enzyme-substrate encounter.

Reviewer #3 (Remarks to the Author):

The identification, cloning, and characterization of a bacterial primase capable of generating DNA primers in standard Mg-containing buffers for in vitro use is a major advance, and in this paper, the authors report a straightforward but very important application of such a technology in facilitating whole-genome amplification by the use of Multiple Displacement Amplification. The authors carefully describe their methods and present data from diverse lines of inquiry, spanning protein structure prediction, careful classical molecular biology experiments, and statistical analysis of coverage and variants in next generation sequencing data. The analyses they present convincingly demonstrate that the formulation of MDA marketed as TruePrime WGA by Sygnis has a much improved coverage profile in comparison to commercial competitor single-cell WGA kits; the allelic dropout statistics (an overall rate difference, at a "typical" coverage depth, of 5% vs 48%!) are particularly impressive.

However, the paper would be improved by some additional analyses that would provide further quantitative rigor to the qualitative views presented in the present manuscript. Exemplary on this point is the question of reproducibility: the Circos plots from replicate cells give only a broad view. We suggest that a sliding window analysis (at multiple window sizes) of coverage in different replicates be presented. Ideally such plots would present not only a whole genome view, but also present exemplary views at finer levels of detail, e.g. within one chromosome or even in a 1 Mb segment. We note that not only coverage could be shown in this way, but also allelic conversion, and that another advantage to showing a genomic track in this way is that you can, on the same plot, also present broadly available other parameters such as GC content, chromatin state, and known ploidy level.

Another class of analysis we would like to see in this paper is a more detailed view of artefactual mutations from the amplification process (e.g. the false positive error rate). The allelic dropout analyses you describe addresses the false negative rate, but the variants you call from your single cells which are not present in the non-amplified material are not only a consequence of the variant caller -- many will be real, but introduced enzymatically, and we would like to see a more thorough

investigation of this. We have given examples of some possible analyses you might try below.

Finally, we would like to comment that certain aspects of the paper seem lacking in polish: e.g. grammatical errors abound, and the figures seem in many cases to be devoid of aesthetic sense, being often taken from canned web-server analyses or Microsoft Office (Figure 5 being a particularly outstanding example).

Introduction --

"contamination from spurious DNA" -- clear what is meant, but sounds a bit awkward. Something like "co-amplification of minute levels of contaminating DNA" may work better.

The penultimate and pre-penultimate paragraphs of the introduction somewhat belabor a simple point: HsPrimPol is capable of generating DNA rather than RNA primers.

"Such convenient capacity was shown to be required to re-prime arrested replication forks during nuclear DNA replication in human cells^{35, 36}, and later confirmed in avian cells⁴⁰." The preceding sentence makes it sound like the DNA nature of the primers is what's responsible for the recruitment of HsPrimPol to arrested replication forks -- but in reality this is a red herring, and the important factor (and an important piece of biological context to emphasize) is the ability of this enzyme to copy over DNA lesions.

Results --

Regarding hsPrimePol -- it is a reasonable argument that a bacterial ortholog will be more straightforward to express, and the Mn-dependence furthers this argument, but if space permits it would be interesting to know a little bit more about why you abandoned the Hsap ortholog -- especially since some of the co-authors seem to have successfully cloned and expressed it in reference 34. Particularly unclear why you believe the Zn-finger domain to be related to stability.

"Database searches" -- such as? More detail is requested.

Please proofread more thoroughly -- there are numerous minor grammatical errors in tense and usage throughout the paper e.g. "conventional AEP-like primases has the three conserved motifs".

I wouldn't call it a scientific requirement, but I think it would be useful to understand and show the evolutionary position of this TthPrimPol a bit more quantitatively -- that is to say, I recommend that you present, in addition to a multiple sequence alignment, a phylogenetic tree relating the sequences you've got in the alignment. A simple ML tree with bootstrap values, calculated on the masked alignment, is probably sufficient.

"Modification of the base preceding the directing TC template sequence had a minor effect on the priming activity of TthPrimPol, in contrast to the strong preference for GTCC shown by human PrimPol³⁴. TthPrimPol was able to initiate DNA primer synthesis also at 30{degree sign}C at multiple sites on a single-stranded circular DNA template (M13mp18) by using both purine and pyrimidine nucleotides to form the initial dimer (Figure 2B), in agreement with a wide template specificity, unlike human PrimPol that largely prefers to make dimers with purine nucleotides³⁴." -- these points alone are enough justification to prefer the Tth ortholog to Hsap, as they greatly increase the diversity of potential initiation sites within the human genome.

It would be interesting, in the context of such WGA experiments, to have direct information on the reproducibility and/or bias in a quantitative sample of primer synthesis sites on a diverse DNA template. The correct assay to do this is not immediately clear, but one immediate question might be: can the primase activity of TthPrimPol incorporate biotin-labeled nucleotides? If so, one could envision a sequencing based assay -- e.g. forming primers in the presence of biotinylated nucleotides, degrading the nucleotides with rSAP, then extending primers with a new polymerase and unlabeled dNTPs, followed by streptavidin-based capture.

I'm not certain that a qualitative assessment of a Circos plot is sufficient to investigate cell-to-cell reproducibility. Maybe a more useful kind of plot would be a look at coverage within windows of various sizes.

Why do you think the chimerism rate is increased in TruePrime WGA? This is somewhat surprising, since there are no differences in the polymerases between this and the commercial MDA methods. Is it simply a matter of adjuvants (e.g. trehalose, see Pan et al 2008 PNAS; 10.1073/pnas.0808028105) that differ between the reactions?

"Therefore, no major parts of the genome appear inaccessible to TruePrime amplification." -- percent coverage is a very coarse metric of this. Again, a k-mer frequency plot would help make your case here.

"From the overall number of SNVs called in common with the non-amplified (reference) sample SNV recovery for the TruePrime amplified genome appeared to be good." I suggest avoiding nebulous, qualitative positive assessments of your own results such as this; let the numbers speak for you.

The difference in allelic dropouts between TruePrime WGA and your commercial competitor are really tremendous -- this is a very impressive aspect of this work. It would be interesting to know if there was also sample-to-sample reproducibility in the variation in ADO (and presumably coverage) among chromosomes. You could also presumably do sliding window analyses on ADO to see how it varies *along* chromosomes, how it co-varies with coverage, and see if there are any correlates with other genomic features, e.g. chromatin state. This is a simple and informative enough analysis that I'd like to request it be included in the resubmitted version of this paper.

Methods --

Your strategy of taking random samples of 12 million reads to assess coverage statistics might be misleading if one of the libraries has a higher PCR duplicate %age than the others. I'd like to request you report this value for all libraries, and if they differ substantially, use Picard to remove duplicates and re-do the coverage analyses on these normalized libraries.

Discussion --

"Indeed, we report a 75% concordance of SNVs called in non-amplified and amplified samples in the range of expected SNV numbers, and most called SNVs outside the common space being likely false positives due to the high sensitivity caller used." -- the false positive SNVs are not only an effect of the variant caller; even a very even coverage WGA process such as yours will see amplification of false positive bases which are incorporated by phi29 early on in the amplification process, which can therefore occur at high VAF. Indeed, it's possible (although unlikely) that the enzymatic priming of the TthPrimPol permits mismatches in the primers that are synthesized, and such mismatches could seed false positive SNVs at a high rate.

A very thorough investigation of the variant calling fidelity and false positive rate of TruePrime WGA would look at a bulk sample which had had subclonal mutations thoroughly quantified using a sensitive existing method e.g. Duplex Seq. Amplification from this bulk sample at single-cell levels of dilution would then show you the rate at which false positive SNVs are generated relative to the known variants from the bulk sequencing. This is probably, however, outside the scope of this paper, which seems only to want to introduce the new method.

This said, we would request that, using the data already generated, a more thorough investigation of false positive variants (artefactual mutations) is attempted. A straightforward way to investigate this would be to focus on a genomic segment which is haploid in Hek293, for which real variants are known from the bulk. The Venn diagrams from 5C&D are, in any case, not a sufficient view of false positive rates.

Figures --

Figure 1A seems to imply that TthPrimPol is actually more processive as a primase in the presence of Mn than Mg.

Caption to 1B : "temptatively" - tentatively

In Figure 3C, it looks as though the WGA products from TruePrime are actually somewhat smaller than from MDA, which might be expected if primers are being generated more frequently than they would anneal in random-priming reactions. It would be good to have a more quantitative view of the sizes of the amplicons prior to fragmentation. I suggest taking some WGA products, treating them with S1 nuclease to remove ssDNA from the debranching process, and then running the products out on an Agilent TapeStation to give a quantitative view of the amplicon sizes (and how they differ from commercial MDA). This will be especially useful to people hoping to use these methods to do longer reads (e.g. in PacBio), or de novo assembly with mate-pair libraries.

Figure 4 -- I think it might be useful to give not only a coverage frequency spectrum but also a kmer-frequency spectrum comparison between the different amplification conditions and the NA control.

Figures 5: ADO (and indeed, coverage) from TruePrime and the commercial competitor should be plotted on the same graph -- indeed this will make the differences between the two much clearer. Furthermore, it is not clear why these parameters are segmented by chromosome - if the goal is only to demonstrate the degree to which they vary, a box plot will suffice for this. If the goal, however, is to investigate variance by linear coordinate -- and ideally, to correlate this variance with other factors, e.g. GC content, heterochromatin state, ploidy level, etc -- then a sliding window plot should be used: one point per chromosome is certainly too coarse.

If you do, as we encourage, present sliding window views (not in the Circos format), we would furthermore request that not only a whole-genome view be presented, but also views at finer levels, e.g. of a single chromosome or chromosome arm, or even within 1 Mb segments. You could for instance cherry pick windows that have the highest between-replicate variability to show.

Miscellaneous --

It seems that the duration of your incubations were relatively short, but it is well known that in MDA, with prolonged amplification (8+ hours), large amounts of high-molecular-weight DNA begin to amplify from no-template controls. It has been suspected that this is related to primer-primer

annealing. It would be interesting to know if your method (which does not have free primers in solution prior to enzymatic activity!) is similar capable of generating no-template amplification products.

I sincerely hope that SYGNIS will eventually market TthPrimPol individually, outside of a kit formulation, as there are many potential uses that a primase could see, e.g. in interrogating single stranded regions of cells whose DNA has not been melted.

REVIEWERS' COMMENTS:

Reviewer #1 (Remarks to the Author):

The manuscript has been greatly revised, with better detail clarification. Single cell whole genome amplification is a fundamental step for single cell genomic studies. The introduction of primase is a new way to improve the amplification. I am glad that, at least based on the data in this manuscript, it works very well.

Reviewer #2 (Remarks to the Author):

The authors have done an excellent job addressing my points. I am, accordingly, enthusiastic about its publication in the Journal.

Reviewer #3 (Remarks to the Author):

Overall, I am happy with the revised manuscripts. While not addressing every point, I feel that the authors have done a good job of addressing my previous concerns.

Answers to Reviewers' comments:

We want to thank all reviewers for their insightful, diligent, and practical comments and suggestions that have helped us to improve the manuscript considerably.

Reviewer #1 (Remarks to the Author):

Picher et al describe a novel whole genome amplification method using TthPrimPol and Phi29. The results presented in the manuscript showed great promise for future single-cell genomic analysis. However, the manuscript was written like a product brochure instead of a scientific publication. Some of my specific comments are as below:

1. In general, the manuscript releases scarce amount of technical details about the TruePrime WGA method and the following analyses. For example, the amplification was first carried out in two separate steps, with different amount of dNTP (Fig 2C). Then for the single cell section, a single step was employed. I understand that this is probably for the protection of the commercial product. However, as a scientific publication, the authors should try their best to facilitate the replication of their results. Given that previous MDA and MALBAC papers all described very detailed protocol, I suggest the authors follow the academic traditions.

As requested by the reviewer, the new amplification methods (in single step) based on the TruePrime strategy are now described in more detail in the Materials and Methods section. We think that we have significantly improved both the methods and result sections to include all necessary details of the experiments which should allow replication of the experiments.

We noticed that we need to clarify that the experiment described in Figure 2C (now Figure 2d) was designed to evaluate the compatibility between TthPrimPol and ϕ 29DNApol. The experiment interrogates about the size of the primers made by PrimPol that can be efficiently extended by ϕ 29DNApol. For that, we designed an experiment in two steps: PULSE step (therefore using a very low concentration of dGTP) with only TthPrimPol (at different enzyme/DNA ratios) to generate labeled primers; CHASE step (using a moderate dNTP concentration (10 μ M), lower than that used for a WGA method), by providing ϕ 29DNApol, to evaluate the extension of the labeled primers.

To clarify both the objective and the technical details of this experiment, a new section (Primase/Polymerase coupled assays) has been included in the revised Methods section. Additionally, both the main text and legend to Figure 2C (now Fig 2d) have been appropriately updated.

2. The single-cell section needs some improvement. First, please list the details about your competitor R control. Again, this is not a product brochure. Second, please add some more comparisons. It is now well known that MDA gets the amplification bias from excess level of amplification. When amplification is limited to \sim 1000 folds, the bias is greatly improved. Following this direction, many studies have shown great results using certain kinds of microfluidics. Since we do not need micrograms of DNA for single-cell sequencing, and MDA can do better when getting just nano grams of DNA, it is not the best comparison I can expect for

Fig 3 and 4. A direct comparison with MALBAC is necessary. 70% coverage stated on p8 is not what they claimed in the original publication.

We have given the requested details about competitor R (REPLI-g sc from Qiagen) in both the Results and Methods sections. Concerning the comparisons, our primary intent here was to set our method apart from the related “traditional” primer-mediated MDA which is why we have chosen only MDA methods as comparison. As requested, we have now added a comparison with MALBAC.

We agree with the reviewer that shorter reaction times that result in lower yields have been shown to be able to reduce bias, and that DNA amounts in the 100 ng range can be sequenced by several labs using low input library protocols. However, a large proportion of sequencing facilities and commercial sequencing services still require DNA at higher concentrations, and the technology should be broadly applicable which is why we target a μ g range of amplification in this first publication that generates sufficient DNA for all users and most purposes.

We have changed the sentence on page 8 discussing coverage breadth to: “Breadth of coverage is a known strength of MDA-based protocols including the hybrid MALBAC method (5, 21, 50) as opposed to purely PCR-based methods (DOP-PCR has only ~10% coverage breadth (3)), and reaches ranges of over 90% genome coverage (51).”

We have performed a MALBAC amplification of single HEK293 cells, and added this to our comparisons.

3. It is not clear to me why the expected coverages in Fig 4A are different across different reactions, given that exact 12M reads were used. If the read quality was the case, the authors should describe.

We apologize for this omission. Indeed, the fraction of mapped reads was different between the samples. We have revised the main text accordingly. The original figure was replaced by a much more informative set of coverage saturation graphs for both 1x and 10 x coverage including a graph on Poisson deviation for 1x coverage (Figure 4d in the revised version).

4. Fig 4C should have Y-axis labels. It is not even clear whether it is linear or log. Same for Fig 5A.

We have added a legend with a scale for the coverage data, the scale is linear, and reaches from 0 to 2500 reads per bin. We have added more details in the figure legend.

5. Lower coverage on chr19 and 22 suggests GC effect on priming. chr19 and 22 have the highest GC contents across all human chromosomes. Actually the coverages were also lower for other chromosomes with high GC content. If this effect is true, I would speculate similar bias within each chromosome, such as promotor regions. Since the authors did not release finer alignment results, it is difficult to tell for now.

As proposed by the reviewer, we have looked deeper into the differences in chromosomal coverage breadth. In a multiple regression model using the coverage breadth pattern from the non-amplified sample and chromosomal GC-content we see that both have a significant effect on the coverage

pattern observed in the TruePrime-amplified sample. The effect of coverage from the non-amplified sample is however dominant, as can be appreciated from the graph (Figure 4e). The difference in chromosomal breadth of coverage in the non-amplified case originates most likely from regions that are difficult to access with the Illumina sequencing technology. There is no influence of chromosomal GC content in the non-amplified case ($p=0.12$). Interestingly, both the TruePrime- and commercial RP-MDA-amplified sample behave exactly the same with regard to the influence of GC content ($R^2=0.38$; $p=0.0017$; $R^2=0.44$, $p=0.0006$) (Supplementary figure 5d). This implies that not the priming mechanism but $\phi 29$ DNApol is responsible for this GC effect. An adequate description has been added to the main text, and complemented with an additional supplementary figure (new Supplementary figure 5). It is important to note that the behavior of the non-amplified material in the Illumina protocol is the main influential covariate for the chromosome coverage breadth behavior of the amplified samples.

6. What is the bin size for each dot in Fig 5B.

Ginkgo uses variable bin sizes to optimally distribute reads (Garvin 2015 Nat Methods). In our case the mean bin size was around 500 kb (Ginkgo analyses are now Supplementary figure 8). We have now added CNV analyses using FREEC (Boeva 2011 Bioinformatics) with a 10-fold higher bin resolution (50 kb fixed bin sizes) (Supplementary figure 7). The CNV analyses were moved to the supplementary section due to size restrictions in figure numbers. Both analyses support our notion of excellent CNV calling using TruePrime amplification.

7. Regarding the chimera formation, as the authors stated, it's from the strand switching during displacement process. The authors didn't show data from MDA, and I don't expect improvement over MDA by changing the priming mechanism. Whether 2-3% increase is a big number should be left for readers' judgement.

We have deleted all evaluative comments in this paragraph. As the reviewer suspected, the REPLIG kit (RP-MDA) shows the same range of broken read pairs (5%) as the TruePrime sample, implying a 2.5% rate of chimera formation.

8. What is the definition of novel SNP on P7? If the authors think high false positive rate was due to software sensitivity, they should adjust the parameters. 25% false positive SNV calling rate is disturbing. 19x sequencing coverage is generally not enough for SNV calling. If the authors can not confidently identify SNV from the bulk sample, I can not trust the analyses on p7. The authors should describe the details about ADO calculation, i.e. which loci were used from the bulk and the single-cell samples.

The intersection of SNVs was done with VCAT v2.2. from Illumina BaseSpace. An SNV is called as "novel" if no entry was found in dbSNP137. The true novelty rate is expected to be in the 5% range for human germline variants in an average human genome.

We have now increased the stringency for the CLCBIO low frequency variant caller, and added a number of other variant callers (Samtools/Bcftools, ISAAC, Varscan2) to strengthen our conclusions regarding the high SNV recovery rate in TruePrime-amplified cells (Table 2 in the revised version. A sequencing depth of 20x gives results that are very close to a 30x coverage (e.g. Cheng 2014 Bioinformatics; Wall 2014 Genome Res; http://www.illumina.com/Documents/products/technotes/technote_snp_caller_sequencing.pdf).

We believe that this depth is in any case sufficient for the purpose of this analysis, to demonstrate high similarity of SNV detectability between the non-amplified sample and the TruePrime-amplified cell.

The ADO rates were determined for the results of all callers/settings now by calculating the conversion rate of heterozygous SNVs in the bulk DNA to homozygous SNVs in the amplified samples over the whole genome, and assuming an equal number of homozygote conversions that are not detected in the SNV overlap due to conversion to the reference allele. Our conclusion of a low ADO in the TruePrime samples were confirmed, although, as already predicted in the discussion of the previous manuscript version, the het -> hom conversion rates vary considerably in dependence of the caller (from 0.73% (ISAAC) to 8.3% (varscan2)).

A detailed method section on the variant callers used and the ADO calling was added to the revised version. A large number of other informative measures has been added to the SNV analyses (see also comments to reviewer 3) (Table 2).

In general, the technology presented in the manuscript is very promising and worth following. However, the manuscript was well written to demonstrate the technology.

We thank the reviewer for this encouraging opinion on the technology.

Reviewer #2 (Remarks to the Author):

In this manuscript, the authors have cloned and characterized the *Thermus thermophilus* PrimPol enzyme and further created a novel primer-free WGA method based on this enzyme for single cell genome amplification. Overall, the manuscript is well written and the conclusions drawn from the data are appropriate.

We appreciate the positive comments about the manuscript and the relevance of our conclusions.

However, a few specific and important points must be addressed.

Specific points: 1. A seq ladder should be included for all PrimPol assays to deduce the exact lengths of the products.

As requested, we have included all possible size markers for the experiments described in Figure 2. Given the combinations of nucleotides used (using different bases and sugars), and the multiple initiation sites on M13 DNA, it is not possible to provide a precise sequence ladder valid for all

lanes. However, and based on multiple size markers with different oligonucleotides, we have modified Figure 2 (parts a to d) indicating an approximate size interval.

2. Fig 2A and B, to further support the wide substrate specificity, the substrates with different sequence from the one used in 2A should be tested.

As requested by the reviewer, we have included an additional experiment in part a of Figure 2 (now appearing as a right panel), showing the effect of single nucleotide changes in the template sequence proximal to the priming site. Main text, figure legend and Methods sections have been modified accordingly.

3. Fig 2C, the two gels of PULSE and CHASE are not well aligned and labeled. Do the highly elongated primers stay in the loading wells? why there are also small amounts of such long products in the PULSE stage (left gel)? Why there are more TthPrimPol-created primers (7-9base) left unextended in the decreased amounts of TthPrimPol?

Figure 2C (now Figure 2d) has been revised, and size markers have been added to both parts. There are some residual labeled primers that are not completely elongated during the chase step, the most prominent being the 9-mer primer; Note that in the left panel (Pulse) the main products are 7, 8 and 9-mers.

4. For the processivity, a more stringent assay (e.g., heparin-based) should be used to ensure single enzyme-substrate encounter.

We fully agree with the reviewer. Such an assay is more frequently used to estimate processivity vs distributivity in nucleic acid synthesizing enzymes. We have performed the requested experiment in the presence of heparin, now presented in Figure 2c, as the first indication supporting the processive synthesis of primers up to 9 nts. Main text, figure legend and Methods sections have been modified accordingly.

Reviewer #3 (Remarks to the Author):

The identification, cloning, and characterization of a bacterial primase capable of generating DNA primers in standard Mg-containing buffers for in vitro use is a major advance, and in this paper, the authors report a straightforward but very important application of such a technology in facilitating whole-genome amplification by the use of Multiple Displacement Amplification. The authors carefully describe their methods and present data from diverse lines of inquiry, spanning protein structure prediction, careful classical molecular biology experiments, and statistical

analysis of coverage and variants in next generation sequencing data. The analyses they present convincingly demonstrate that the formulation of MDA marketed as TruePrime WGA by Sygnis has a much improved coverage profile in comparison to commercial competitor single-cell WGA kits; the allelic dropout statistics (an overall rate difference, at a "typical" coverage depth, of 5% vs 48%!) are particularly impressive.

We thank the reviewer for the positive and encouraging comments about the relevance of our data.

However, the paper would be improved by some additional analyses that would provide further quantitative rigor to the qualitative views presented in the present manuscript. Exemplary on this point is the question of reproducibility: the Circos plots from replicate cells give only a broad view. We suggest that a sliding window analysis (at multiple window sizes) of coverage in different replicates be presented. Ideally such plots would present not only a whole genome view, but also present exemplary views at finer levels of detail, e.g. within one chromosome or even in a 1 Mb segment. We note that not only coverage could be shown in this way, but also allelic conversion, and that another advantage to showing a genomic track in this way is that you can, on the same plot, also present broadly available other parameters such as GC content, chromatin state, and known ploidy level.

As suggested by the reviewer we have provided close-up views of the reproducibility experiment (Figure 6c and Supplementary figure 6). We have also added a correlation matrix for the read coverage in bins of 100 kb for the 4 single cell amplifications that demonstrate the high similarity in the coverage pattern (Figure 6b).

We have also generated Circos plots showing ADO rates together with other genomic features (Figure 7).

Another class of analysis we would like to see in this paper is a more detailed view of artefactual mutations from the amplification process (e.g. the false positive error rate). The allelic dropout analyses you describe addresses the false negative rate, but the variants you call from your single cells which are not present in the non-amplified material are not only a consequence of the variant caller -- many will be real, but introduced enzymatically, and we would like to see a more thorough investigation of this. We have given examples of some possible analyses you might try below.

We have strengthened the whole SNV calling, ADO, and FPR part with a comparison of outcomes from 4 different callers, and by using more stringent settings for the caller used in the previous version of the paper. As suggested by the reviewer, we have analyzed the false positive rate (FPR) in our SNV analyses (Table 2). This ranges from 0.88 to 3.66% depending on the caller used.

We looked at two other possible measures of false positives: The relative increase in novelty rate (SNVs not found in dbSNP138) of the TruePrime-amplified sample versus the non-amplified sample ranges from 2.48% (CLC lowfreq stringent), 3.94% (VarScan2) to 13.26% (Isaac) and 36% (samtools). Also, we determined the conversion of homozygote alleles in non-amplified to heterozygote alleles in the amplified samples. This ranges from 0.25% to 5.61% with an apparent problem in the samtools/bcftools caller which detected a very high rate of 20%. This was also done

for chromosome 18 only, a haploid chromosome in our cell line as suggested by this reviewer (see question below).

We have added a new section on this in the revised manuscript. The differences seen between the different callers highlight that measures of false positives, as with ADO calling, is highly dependent on the SNV caller and needs to be interpreted having this dependency in mind.

Finally, we would like to comment that certain aspects of the paper seem lacking in polish: e.g. grammatical errors abound, and the figures seem in many cases to be devoid of aesthetic sense, being often taken from canned web-server analyses or Microsoft Office (Figure 5 being a particularly outstanding example).

We have considerably improved the quality and resolution of all figures, especially for the bioinformatic analyses.

Introduction --

"contamination from spurious DNA" -- clear what is meant, but sounds a bit awkward. Something like "co-amplification of minute levels of contaminating DNA" may work better.

OK

The penultimate and pre-penultimate paragraphs of the introduction somewhat belabor a simple point: HsPrimPol is capable of generating DNA rather than RNA primers.

"Such convenient capacity was shown to be required to re-prime arrested replication forks during nuclear DNA replication in human cells^{35, 36}, and later confirmed in avian cells⁴⁰."

The preceding sentence makes it sound like the DNA nature of the primers is what's responsible for the recruitment of HsPrimPol to arrested replication forks -- but in reality this is a red herring, and the important factor (and an important piece of biological context to emphasize) is the ability of this enzyme to copy over DNA lesions.

We agree with the reviewer that PrimPol is also competent for directly bypassing (copying) lesions, as a conventional TLS DNA polymerase. However, the existing *in vivo* evidence demonstrates that the primase function is the one absolutely required to re-start stalled replication forks arising upon replicative stress induced by UV light or dNTP depletion (HU treatment). For the moment, there is no convincing data demonstrating the *in vivo* impact of PrimPol copying over DNA lesions. Recently, a new publication supports the relevance of human PrimPol to tolerate (via re-priming) G4 Quadruplexes in vertebrate cells (Schiavone et al., 2016 Mol. Cell), which has been included in the Reference section of the revised version

Regarding hsPrimePol -- it is a reasonable argument that a bacterial ortholog will be more straightforward to express, and the Mn-dependence furthers this argument, but if space permits it would be interesting to know a little bit more about why you abandoned the Hsap ortholog -- especially since some of the co-authors seem to have successfully cloned and expressed it in reference 34. Particularly unclear why you believe the Zn-finger domain to be related to stability.

The bacterial enzyme is more stable and robust, probably as a consequence of its thermostability. That allows a more exhaustive purification, particularly free of host cell DNA, as required for the purpose of its use in amplification. Conversely, the human enzyme is more labile, and although it has been successfully purified in active form (Garcia-Gomez, 2013, Mol Cell), the stability of the final fraction is not too high. That implies the need to carry out fresh purifications, particularly when you need a direct comparison (i.e. wild-type versus mutant versions). The assumption that the Zn-finger is part of the stability problems stems from the higher stability observed when preparing a Zn-finger deleted mutant of HsPrimPol (as described in Mourón et al. 2013 NSMB). Unfortunately for the purpose of its use in amplification, that Zn finger-deleted mutant is devoid of primase activity.

"Database searches" -- such as? More detail is requested.

Agreed. Now we clearly indicate that the database search was performed using the BLAST program, on the Protein database from NCBI.

Please proofread more thoroughly -- there are numerous minor grammatical errors in tense and usage throughout the paper e.g. "conventional AEP-like primases has the three conserved motifs".

We apologize for the errors. The new version of the manuscript has been subjected to a rigorous proofreading.

I wouldn't call it a scientific requirement, but I think it would be useful to understand and show the evolutionary position of this TthPrimPol a bit more quantitatively -- that is to say, I recommend that you present, in addition to a multiple sequence alignment, a phylogenetic tree relating the sequences you've got in the alignment. A simple ML tree with bootstrap values, calculated on the masked alignment, is probably sufficient.

The evolutionary position of PrimPol proteins in general, as part of the AEP superfamily, has been extensively studied by the Koonin group (Iyer et al., 2005, NAR). However, in that study *HsPrimPol* was predicted as a new primase belonging to the Herpes virus-like subclade of primases, but not as a PrimPol. This prediction failure was due to the complexity of this

superfamily, constituted by 13 distinct families from eukaryotic, archaeal, bacterial, viral and plasmidic origins (see below Figure 3 from Iyer, 2005, NAR, reproduced here just for revision).

Figure 3. Inferred evolutionary history of the AEP superfamily. The overall topology of the phylogram was derived using synapomorphies and clustering based on DALI Z-scores. Synapomorphies that unify a set of lineages are indicated next to the filled yellow circles. The ellipses indicate large assemblies within which individual lineages show a generic relationship. Broken lines indicate an uncertainty with respect to the exact point of origin of a lineage. Archaeal and eukaryotic (including viral) branches are colored blue, bacterial branches are colored green, branches that include predominantly proteins from plasmids, phages and mobile elements are colored red. Ancestral branches and branches outside the AEP superfamily are in black. The phyletic distribution is shown in brackets: B, Bacteria; A, Archaea; E, Eukaryotes; V, Viruses; > represents a proposed lateral transfer.

Because of that precedent, the alignment that we present in Figure 1b only includes the closest bacterial orthologues, and some more evolutionary distant members of the superfamily, two of them already characterized as PrimPols based on *in vitro* enzymatic analysis (pRN1; BcMCM). However, considering the reviewer's suggestion, aimed to understand and show the evolutionary position of this *Tth*PrimPol, we carried out a phylogenetic analysis of the first 90 bacterial close relatives of *Tth*PrimPol, that can be considered to be PrimPol orthologues. This has been now included in the revised version as new Supplementary Figure 1.

"Modification of the base preceding the directing TC template sequence had a minor effect on the priming activity of *Tth*PrimPol, in contrast to the strong preference for GTCC shown by human PrimPol34. *Tth*PrimPol was able to initiate DNA primer synthesis also at 30°C at multiple sites on a single-stranded circular DNA template (M13mp18) by using both purine and

pyrimidine nucleotides to form the initial dimer (Figure 2B), in agreement with a wide template specificity, unlike human PrimPol that largely prefers to make dimers with purine nucleotides³⁴." -- these points alone are enough justification to prefer the Tth ortholog to Hsap, as they greatly increase the diversity of potential initiation sites within the human genome.

We fully agree with the reviewer. We have slightly modified the text to emphasize this as an advantage for the method described here.

It would be interesting, in the context of such WGA experiments, to have direct information on the reproducibility and/or bias in a quantitative sample of primer synthesis sites on a diverse DNA template. The correct assay to do this is not immediately clear, but one immediate question might be: can the primase activity of TthPrimPol incorporate biotin-labeled nucleotides? If so, one could envision a sequencing based assay -- e.g. forming primers in the presence of biotinylated nucleotides, degrading the nucleotides with rSAP, then extending primers with a new polymerase and unlabeled dNTPs, followed by streptavidin-based capture.

We thank the reviewer for these suggestions and experimental ideas. We are considering such a detailed analysis, but believe that this will take many months to perform and is outside the scope of the present paper.

I'm not certain that a qualitative assessment of a Circos plot is sufficient to investigate cell-to-cell reproducibility. Maybe a more useful kind of plot would be a look at coverage within windows of various sizes.

As suggested by the reviewer we have added close-up views of coverage for the reproducibility experiment (Supplementary figure 6). In addition, we have added a cross-correlation of binned coverage depths (Figure 6b).

Why do you think the chimerism rate is increased in TruePrime WGA? This is somewhat surprising, since there are no differences in the polymerases between this and the commercial MDA methods. Is it simply a matter of adjuvants (e.g. trehalose, see Pan et al 2008 PNAS; 10.1073/pnas.0808028105) that differ between the reactions?

This is probably a misunderstanding. We originally stated only the difference between non-amplified and TruePrime in terms of broken read pairs due to inversion or wrong distance. The assumed fraction of chimeras due to the amplification process is 2-3% in both TruePrime and the commercial RP-MDA method. We concur with the reviewer in assuming that the main culprit in this is ϕ 29 DNA polymerase. We have now rephrased this section to clarify this (page 6) (see also comment from reviewer 1).

"Therefore, no major parts of the genome appear inaccessible to TruePrime amplification." -- percent coverage is a very coarse metric of this. Again, a k-mer frequency plot would help make your case here.

We have performed K-mer analysis as suggested. K-mers were calculated with jellyfish, K-mer size was set to 19 as recommended by Kelley et al. 2010, Genome Biology. K-mer characteristics are quite similar between the non-amplified and TruePrime-amplified sample with an estimation of the single copy region of 0.79 in the non-amplified and 0.81 in the TruePrime-amplified sample, but the bimodal distribution in the commercial RP-MDA sample (and MALBAC) is completely lost. We have included a figure (Figure 5) on this analysis and added sections in result and methods sections.

"From the overall number of SNVs called in common with the non-amplified (reference) sample SNV recovery for the TruePrime amplified genome appeared to be good." I suggest avoiding nebulous, qualitative positive assessments of your own results such as this; let the numbers speak for you.

We have tried to omit all evaluative statements in the result section in the revised version, and deleted the above sentence.

The difference in allelic dropouts between TruePrime WGA and your commercial competitor are really tremendous -- this is a very impressive aspect of this work. It would be interesting to know if there was also sample-to-sample reproducibility in the variation in ADO (and presumably coverage) among chromosomes. You could also presumably do sliding window analyses on ADO to see how it varies *along* chromosomes, how it co-varies with coverage, and see if there are any correlates with other genomic features, e.g. chromatin state. This is a simple and informative enough analysis that I'd like to request it be included in the resubmitted version of this paper.

We thank the reviewer for those encouraging comments. We have added a figure (Figure 7) showing distribution of ADOs across the genome together with coverage and GC content. However, we cannot answer the question regarding the reproducibility of ADO patterns in different samples at present as the amplifications demonstrating reproducibility of coverage were only sequenced to low depth.

Methods

--

Your strategy of taking random samples of 12 million reads to assess coverage statistics might be misleading if one of the libraries has a higher PCR duplicate %age than the others. I'd like to request you report this value for all libraries, and if they differ substantially, use Picard to remove duplicates and re-do the coverage analyses on these normalized libraries.

We thank the reviewer for this observation. Indeed, PCR duplicates will be mapped successfully but not count towards coverage breadth. The duplicate rates were generally very low (from 1.23% in the TruePrime sample to 1.79% in the non-amplified sample and 0.23% in the MALBAC sample) except for the generic random-primed MDA sample that had a duplicate rate of 9.53% (and is not followed further in the deep sequencing experiments). We have recalculated the

expected coverage breadth by correcting for those numbers for the 12 mio input calculation in the text. The deviations observed to expected have slightly changed to 9.17% for non-amplified DNA, 13.15% for TruePrime, 34.83% for commercial random-primed MDA and 30.73% for generic random-primed MDA. The new numbers do not change the conclusions from this analysis. We have added more informative coverage saturation graphs now (Figure 4d).

Discussion

--

"Indeed, we report a 75% concordance of SNVs called in non-amplified and amplified samples in the range of expected SNV numbers, and most called SNVs outside the common space being likely false positives due to the high sensitivity caller used." -- the false positive SNVs are not only an effect of the variant caller; even a very even coverage WGA process such as yours will see amplification of false positive bases which are incorporated by phi29 early on in the amplification process, which can therefore occur at high VAF. Indeed, it's possible (although unlikely) that the enzymatic priming of the TthPrimPol permits mismatches in the primers that are synthesized, and such mismatches could seed false positive SNVs at a high rate.

A very thorough investigation of the variant calling fidelity and false positive rate of TruePrime WGA would look at a bulk sample which had had subclonal mutations thoroughly quantified using a sensitive existing method e.g. Duplex Seq. Amplification from this bulk sample at single-cell levels of dilution would then show you the rate at which false positive SNVs are generated relative to the known variants from the bulk sequencing. This is probably, however, outside the scope of this paper, which seems only to want to introduce the new method.

This said, we would request that, using the data already generated, a more thorough investigation of false positive variants (artefactual mutations) is attempted. A straightforward way to investigate this would be to focus on a genomic segment which is haploid in Hek293, for which real variants are known from the bulk. The Venn diagrams from 5C&D are, in any case, not a sufficient view of false positive rates.

In addition to the parameters mentioned in a previous remark to address the question of false positives possibly generated by the amplification technique (total hom -> het conversion, FPR, novelty rates), we have followed the reviewer's suggestion and determined the hom -> het conversion rate for chromosome 18 which is haploid in our cell line (see CNV profiles) (Table 2).

Figures

--

Figure 1A seems to imply that TthPrimPol is actually more processive as a primase in the presence of Mn than Mg.

We did not perform a direct measurement of primase processivity as a function of the metal activator (Mn vs Mg). From the figure alluded to by the reviewer (Fig 2a), such a conclusion cannot be firmly extracted. In response to reviewer 2, we have carried out an additional experiment

(based on the use of heparin to capture dissociated PrimPol molecules) to support that the primers made in the presence of magnesium are processively synthesized up to 9 nt. These are the conditions that are relevant in this paper, given the negative fidelity impact on ϕ 29DNA polymerase when using manganese ions.

Caption to 1B : "temptatively" – tentatively

OK

In Figure 3C, it looks as though the WGA products from TruePrime are actually somewhat smaller than from MDA, which might be expected if primers are being generated more frequently than they would anneal in random-priming reactions. It would be good to have a more quantitative view of the sizes of the amplicons prior to fragmentation. I suggest taking some WGA products, treating them with S1 nuclease to remove ssDNA from the debranching process, and then running the products out on an Agilent TapeStation to give a quantitative view of the amplicon sizes (and how they differ from commercial MDA). This will be especially useful to people hoping to use these methods to do longer reads (e.g. in PacBio), or de novo assembly with mate-pair libraries.

We have added a quantification of fragment sizes generated by TruePrime, RP-MDA, and MALBAC (Supplementary figure 3).

Figure 4 -- I think it might be useful to give not only a coverage frequency spectrum but also a kmer-frequency spectrum comparison between the different amplification conditions and the NA control.

We have included a k-mer frequency analysis now (Figure 5; see also comment above).

Figures 5: ADO (and indeed, coverage) from TruePrime and the commercial competitor should be plotted on the same graph -- indeed this will make the differences between the two much clearer. Furthermore, it is not clear why these parameters are segmented by chromosome - if the goal is only to demonstrate the degree to which they vary, a box plot will suffice for this. If the goal, however, is to investigate variance by linear coordinate -- and ideally, to correlate this variance with other factors, e.g. GC content, heterochromatin state, ploidy level, etc -- then a sliding window plot should be used: one point per chromosome is certainly too coarse.

We have added a number of figures in the revised version to give more appropriate views of coverage patterns, ADO distributions, and CNV predictions (e.g. Figure 7, Supplementary figures 4,6,7,8).

If you do, as we encourage, present sliding window views (not in the Circos format), we would furthermore request that not only a whole-genome view be presented, but also views at finer levels,

e.g. of a single chromosome or chromosome arm, or even within 1 Mb segments. You could for instance cherry pick windows that have the highest between-replicate variability to show.

Done (Supplementary figures 4 and 6).

Miscellaneous

--

It seems that the duration of your incubations were relatively short, but it is well known that in MDA, with prolonged amplification (8+ hours), large amounts of high-molecular-weight DNA begin to amplify from no-template controls. It has been suspected that this is related to primer-primer annealing. It would be interesting to know if your method (which does not have free primers in solution prior to enzymatic activity!) is similar capable of generating no-template amplification products.

As the reviewer emphasizes, the main advantage of the TruePrime method is the highly reduced propensity to generate primer-derived artefacts, as also pointed out in our manuscript. No background is obtained when using moderate incubation times (up to 3h) of amplification. However, when the amplification reaction is maintained for longer periods some unspecific products can be obtained, probably derived from minute amounts of contaminant DNA and the extreme sensitivity of our method; it is important to point that the presence of an input DNA as low as 1 fg is enough to avoid generation of these products (as shown in Supplementary. figure 2). Alternatively, we cannot discard that *Tth*PrimPol could be able to synthesize un-templated primers of certain length that can later produce primer-derived artefacts of amplification.

I sincerely hope that SYGNIS will eventually market *Tth*PrimPol individually, outside of a kit formulation, as there are many potential uses that a primase could see, e.g. in interrogating single stranded regions of cells whose DNA has not been melted.

We thank the reviewer for this encouragement to identify other potential uses of *Tth*PrimPol outside of a kit. Sygnis is open to discuss any potential interesting applications of *Tth*PrimPol with interested researchers.